# Incentivizing Desirable Effort Profiles in Strategic Classification: The Role of Causality & Uncertainty

**Valia Efthymiou**
MIT ORC
valia554@mit.edu

**Chara Podimata**
MIT Sloan School of Management &
Archimedes/Athena RC, Greece
podimata@mit.edu

**Diptangshu Sen**
Georgia Tech ISyE
dsen30@gatech.edu

**Juba Ziani**
Georgia Tech ISyE
jziani3@gatech.edu

## Abstract

We study strategic classification in binary decision-making settings where agents can modify their features in order to improve their classification outcomes. Importantly, our work considers the causal structure across different features, acknowledging that effort in one feature may affect other features. The main goal of our work is to understand *when and how much agent effort is invested towards desirable features*, and how this is influenced by the deployed classifier, the causal structure of the agent's features, their ability to modify them, and the information available to the agent about the classifier and the feature causal graph. We characterize conditions under which agents with full information about the causal structure and the principal's classifier align with the principal's goals of incentivizing effort mostly in "desirable" features, and identify cases where designing such classifiers (from the principal's side) is still tractable despite general non-convexity. Under incomplete information, we show that uncertainty leads agents to prioritize features with high expected impact and low variance, which may often be misaligned with the principal's goals. Finally, using numerical experiments based on a cardiovascular disease risk study, we illustrate how to incentivize desirable modifications even under uncertainty.

## 1 Introduction

The widespread adoption of automated decision-making systems has brought significant attention to the issue of *strategic classification*—a machine learning setting where individuals modify their features to secure favorable outcomes. This phenomenon is common in many domains: students enroll in preparatory courses to enhance their chances at college admission; job seekers tailor their resumes to align them with AI-based hiring algorithms, and individuals adjust their financial behaviors to improve credit scores. Some of these modifications reflect *genuine efforts* to enhance one's qualifications or financial responsibility (e.g., acquiring new skills or consistently paying off loans), while others *effectively game* the system, (e.g., artificially boosting credit scores by opening new credit lines or strategically targeting specific keywords in algorithmic resume screening).

The distinction between *desirable* and *undesirable* modifications is not always clear-cut. While gaming is typically regarded as problematic, even genuine improvements can vary in how desirable they are. For example, in healthcare, encouraging patients to adopt preventive lifestyle changes (such as improved diet and regular exercise) may be preferable to medical interventions like medication or surgery for conditions such as obesity or hyperlipidemia. This highlights the nuanced nature of

39th Conference on Neural Information Processing Systems (NeurIPS 2025).

strategic classification: interventions that lead to real improvements may still not align with preferred or *desirable* forms of improvement, where desirability is decided by the learner.

Further, a key challenge in strategic classification is that features are often interdependent. That is, modifications to one feature can have cascading effects on others. For example, increasing the number of credit cards an individual holds will also lower their credit utilization percentage, indirectly influencing their credit-worthiness. Similarly, reducing alcohol consumption or improving dietary habits can mitigate multiple health risks, such as obesity, hyperlipidemia, and cardiovascular diseases. These dependencies are best captured using a *causal graph*, a framework that has been explored in a limited amount of prior work [5, 26, 32, 40] in the specific context of strategic classification.

Our work builds upon this causal perspective on strategic classification, investigating how agents respond to decision-making systems and, in particular, when their strategic behavior aligns with desirable modifications. We adopt a framework in which a principal (e.g., a decision-maker or machine learning classifier) deploys a model, and agents (or individuals) strategically adjust their features to maximize their probability of receiving a favorable classification outcome.

**Our contributions.** Our paper makes the following contributions.

*Model.* In Section 2, we introduce our model to study incentivizing desirable efforts in the context of causal strategic classification. We distinguish from previous work in two ways: *i)* we introduce *incomplete information to the study of causality in strategic classification* capturing settings where agents do not know the classifier, causal graph, or both; and *ii)* we introduce a notion of $\beta$-desirability quantifying the extent to which agents invest effort in features deemed desirable by the principal.

*Complete Information.* In Section 3, we focus on agents with full knowledge of the classifier and the causal structure, and we characterize their optimal effort profiles under various effort cost structures. We establish theoretical conditions guaranteeing investment in desirable effort profiles by rational agents. We also demonstrate that finding classifiers that induce desirable behavior is, in general, a non-convex problem. However, we show that when the principal chooses only *one* desirable feature to incentivize—a special case that is particularly important in practice—, the problem of finding good classifiers becomes convex. We also provide a simple "convexification" heuristic for when the number of desirable features is more than one, ensuring that chosen classifiers do *not* incentivize more than a certain amount of undesirable feature effort.

*Incomplete Information.* We extend our analysis to settings where agents lack information about either the classifier or the causal graph (or both) in Section 4; incomplete information is modeled as agents having Gaussian priors about the classifier and the causal graph. We show that in face of uncertainty over both the classifier and the causal graph, investing effort optimally is a non-convex problem for the agent. However, the problem becomes tractable under *partial uncertainty*, and we provide semi-closed-form characterizations of optimal effort profiles in some special settings.

*Case study.* Finally, in Section 5, we complement our theoretical insights in the incomplete information setting with numerical experiments, basing our experimental setup on a medical study from previous work that predicts *risk of cardiovascular disease* (CVDs) in adults. In the process, we provide insights into how to incentivize changes in desirable features under uncertainty.

**Related work.** Strategic classification has received significant attention over the past decade (see e.g., [1, 2, 5, 8, 9, 11, 14, 17, 20, 21, 26, 30, 32, 33, 37, 40, 41, 44]). Among this active area of research, perhaps closest to our work is the work of [26]. Like us, they focus on general causal graphs; however, we highlight several major differences. First, we focus on *classification* settings, while [26] focus on regression and scoring settings. Second, we highlight differences in our agent model, where our agents invest effort to pass the classifier with reasonably high probability, while agents in [26] *always* exert effort to improve their score. Third, we note that our cost model is strictly more general: where [26] focuses on linear costs, our work considers general $\ell_p$-costs[1]. Finally, unlike [26], our study incorporates *incomplete information*, where agents may not fully understand either the causal graph or the deployed classifier.

Our work is also closely related to the literature on "Algorithmic Recourse" [24, 25, 42, 43] which explores how to provide individuals, who receive adverse decisions from machine learning models,

---

[1]Our results show that this choice of cost is important, noting a sharp distinction in agent behavior between the cases of $\ell_1$-cost and $\ell_p$ costs for $p > 1$.

with feedback (in the form of counterfactual explanations or recommendations) for more favorable future outcomes. From a mathematical standpoint, strategic classification and algorithmic recourse can be seen as flip sides of the same picture: recourse has the learner tell the agent what actions to take to improve their outcomes[2], while strategic classification sees agents as acting by themselves decide their own actions based on the classifier. A key difference is that much of the recourse literature aims to find a low-cost path between an agent's current features and features that lead to positive classification, often without asking whether this path is "desirable" nor whether it involves gaming the classifier or investing in true improvements. [28] show, in fact, that standard counterfactual recourse algorithms often lead to undesirable outcomes, while our work explicitly aims to steer away from those. For a detailed survey on recourse, please refer to [23].

For a full discussion of related work, please refer to Appendix A.

## 2  Model

We consider a *binary classification* problem, where there is an interaction between a principal and an agent. Roughly, the principal (aka *learner*), deploys an ML model or classifier, which assigns a binary decision in $\{0, 1\}$ to each agent, based on her *features*. Finally, agents *respond* to the deployed classifier, potentially changing their features to obtain better outcomes, at a cost.

Let $\mathcal{F}$ be the set of features. Each agent $k$ is defined by a feature vector $x_k \in \mathbb{R}^d$ where $|\mathcal{F}| = d$. The set of features $\mathcal{F}$ is partitioned into $\mathcal{D}$ (the set of **desirable** features) and $\mathcal{U}$ (the set of **undesirable** features). Informally, *desirable* features are those that the principal wants to incentivize the agent to change directly; e.g., in the health application of the Introduction, "alcohol consumption" would be a *desirable* feature that the principal (e.g., the agent's primary care physician) would like to see lowered. *Undesirable* features, on the other hand, can be considered as features that we would like to disincentivize agents from modifying directly: e.g., directly intervening to lower an agent's cholesterol level via medication such as statins may be less desirable than promoting lifestyle changes (lower alcohol consumption, improved diet, etc.) that will also lower their cholesterol.

**Causal feature interactions.** We adopt a *causal* perspective where different features can impact each other—i.e., a change in a feature $i$ (e.g., alcohol consumption or diet) that has a causal relationship with feature $j$ (e.g., cholesterol) will also induce a change in feature $j$. The chain of causality between the different features can be captured using a weighted directed graph $\mathcal{G} = (\mathcal{F}, \mathcal{A}, w)$, called the *causal graph*. $\mathcal{A}$ represents the set of directed edges on $\mathcal{G}$, where an edge from features $i$ to $j$ indicates that $i$ is causal for $j$. Finally, $w : \mathcal{A} \to \mathbb{R}$ captures the weights of the edges. We make no assumption on the structure of $\mathcal{G}$, other than the fact that it is a directed *acyclic* graph. [3]

We represent all necessary information about the graph using an *adjacency matrix* $A \in \mathbb{R}^{d \times d}$, where $A_{ij} = w(a_{ij})$ if $a_{ij} \in \mathcal{A}$ and 0 otherwise. If there is an edge $a_{ij} \in \mathcal{G}$, then feature $i \in \mathcal{F}$ *causally affects* feature $j \in \mathcal{F}$ directly. The weight of edge $a_{ij}$, given by $w(a_{ij})$ indicates that if feature $i$ improves by a unit amount, then the value of the downstream feature $j$ will improve by $w(a_{ij})$[4].

**Contribution matrix.** We define the contribution matrix $\mathbb{C} \in \mathbb{R}^{d \times d}$ for causal graph $\mathcal{G}$ as:

$$\mathbb{C}_{ii} = 1 \quad \forall\, i \in [d], \quad \text{and} \quad \mathbb{C}_{ij} = \sum_{p \in \mathcal{P}_{ij}} \omega(p) \quad \forall\, i, j \in [d],\ i \neq j,$$

where $\mathcal{P}_{ij}$ is the set of all directed paths from node $i$ to node $j$ on $\mathcal{G}$ and $\omega(p)$ is the weight of path $p \in \mathcal{P}_{ij}$ with $\omega(p) = \prod_{a \in \mathcal{A}, a \subset p} w(a)$.

In causal graphs, feature $i$ may affect another feature $j$ not just directly (in which case there would be an edge of non-zero weight from $i$ to $j$), but also *indirectly* through other features; if there is a directed path from feature $i$ to feature $j$ through intermediary features $i_1, \ldots, i_k$, where $i \to i_1 \to i_2 \to \ldots \to i_k \to j$[5], then this can be encoded by the contribution matrix (Figure 1). Given the

---

[2]Agents may or may not follow the learner's recommendations, depending on how well their incentives are aligned with said recommendations.

[3]This is standard in the causal strategic classification [26, 32].

[4]Throughout the paper, we assume that causal relationships are linear. This is another common assumption in the literature [26, 40].

[5]$x \to y$ indicates that $x$ is directly causal for $y$.

adjacency matrix $A$, we can also show that the contribution matrix $\mathbb{C}$ can be computed efficiently (see Appendix E).

## 2.1 Principal - Agent Interaction

The principal deploys a *linear classifier* $h_0 \in \mathbb{R}^d$. Under this classifier, an agent with feature vector $x \in \mathbb{R}^d$ is assigned a score of $s(x) = h_0^\top x$. The classification decision $y$ for said agent is given by $y(x) = \mathbf{1}\left[s(x) \geq \tau\right]$ for a pre-determined threshold $\tau \in \mathbb{R}$.

**Agent Information Structure.** We assume that the agent has Gaussian priors $\Pi_h := \mathcal{N}(\mu_h, \Sigma_h)$ (where $\mu_h \in \mathbb{R}^{|\mathcal{F}|}, \Sigma_h \in \mathbb{R}^{|\mathcal{F}| \times |\mathcal{F}|}$) over the deployed classifier $h_0$ and $\Pi_\mathbb{C} := \mathcal{N}(\mu_w, \Sigma_w)$ (where $\mu_w \in \mathbb{R}^{|\mathcal{A}|}, \Sigma_w \in \mathbb{R}^{|\mathcal{A}| \times |\mathcal{A}|}$) over the edge weights of the causal graph $\mathcal{G}$[6]. The agent knows the topology of the causal graph. We explore two kinds of information structures:

1. The *Complete Information setting* (Section 3), where the agent fully knows the classifier $h_0$, i.e, $\mu_h = h_0$ and $\Sigma_h = \mathbf{0}$ and the weights of all edges of $\mathcal{G}$, i.e., $\mu_w = w$ and $\Sigma_w = \mathbf{0}$.
2. The *Incomplete Information setting* (Section 4), where *i)* there is uncertainty over the principal's classifier $h_0$, i.e., $\mu_h$ may differ from $h_0$ (bias) and $\Sigma_h \neq \mathbf{0}$ (variance), and/or *ii)* there is uncertainty over the edge weights of the causal graph $\mathcal{G}$, i.e., $\mu_w$ may differ from $w$ (bias) and $\Sigma_w \neq \mathbf{0}$ (variance).

If $y(x) = 0$, the agent also knows the amount $\alpha > 0$ by which she fell short of passing the classifier; e.g., in a loan approval setting, an agent may know their current credit score and be told the threshold credit score that the bank uses to decide who gets approved for a loan.

**Agent Best Response.** The agent wishes to obtain a positive classification outcome, i.e. $y(x) = 1$; she attempts to *modify* her feature vector $x$ by investing some *exogenous effort* $e \in \mathbb{R}^{|\mathcal{F}|}$, which we call the agent's *exogenous effort profile*. Importantly, exerting exogeneous effort on a subset of the features can also lead to other features (particularly those on which no effort was exerted) to change *endogenously*, due to causality. We call this phenomenon *induced* or *endogenous feature change*.

Exerting effort comes at a cost, modeled through a *cost function* $\mathsf{Cost} : \mathbb{R}^d \to \mathbb{R}_{\geq 0}$, where $\mathsf{Cost}(e)$ is the cost incurred for effort $e$. We mainly focus on (weighted) $\ell_p$-norm cost functions for all $p \geq 1$:

$$\mathsf{Cost}(e) = \left(\sum_{f \in \mathcal{F}} c_f \left|e_f\right|^p\right)^{1/p}, \text{ where } c_f > 0 \ \forall f \in \mathcal{F}, \tag{1}$$

where $c_f$ represents a cost multiplier associated with investing unit exogenous effort into feature $f$.

Let $x'(e)$ be the agent's modified feature vector after investing effort $e$, and let $\Delta x(e) = x'(e) - x$ be the net change in features due to $e$. Given the contribution matrix $\mathbb{C}$ and exogenous effort $e$, $\Delta x(e)$ is given by $\Delta x(e) = \mathbb{C}^\top e$. Hence, the agent's objective is to choose their optimal effort profile $e^\star$ that ensures that $y(x'(e^\star)) = 1$ with probability at least $1 - \delta$, while incurring the minimum possible cost. We call the effort profile $e^\star(\Pi_h, \Pi_\mathbb{C})$ the agent's *best response* to priors $(\Pi_h, \Pi_\mathbb{C})$. Formally:

$$e^\star(\Pi_h, \Pi_\mathbb{C}) = \arg\min_e \quad \mathsf{Cost}(e) \quad \text{s.t.} \quad \mathbb{P}_{h \sim \Pi_h, \mathbb{C} \sim \Pi_\mathbb{C}}\left[h^\top \mathbb{C}^\top e \geq \alpha\right] \geq 1 - \delta. \tag{2}$$

Note that the constraint ensures that an agent passes the classifier with probability at least $1 - \delta$, with respect to their prior on the causal graph and the classifier. In particular, if they exert effort profile $e$, their features change by $\mathbb{C}^\top e$, so their score changes by $h^\top \mathbb{C}^\top e$, and they have to improve their score by at least $\alpha$ to pass the classifier.

## 2.2 Incentivizing Effort towards Desirable Features

We are interested in the *properties* of the effort profile that the agent exerts as a result of best-responding to the principal's classifier, and in particular understanding the amount of effort they exert towards desirable features in set $\mathcal{D}$ and undesirable features $\mathcal{U}$. The goal is to incentivize effort

---

[6]Gaussian priors are frequently used to model incomplete information [13, 27].

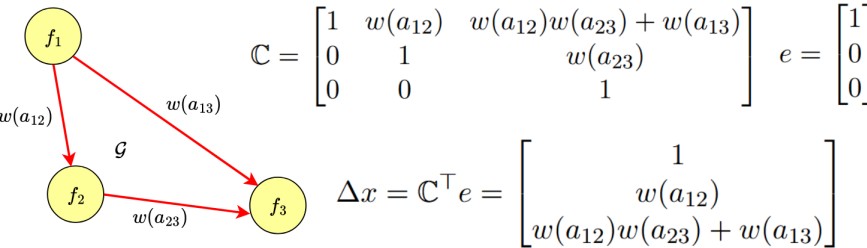

Figure 1: Example of a causal graph $\mathcal{G}$ with $|\mathcal{F}| = 3$. $f_1$ directly affects $f_2$ and $f_3$ and $f_2$ directly affects $f_3$. $f_1$ also indirectly affects $f_3$ through the path $1 \to 2 \to 3$.

towards desirable features and away from undesirable features, i.e. to understand *when is it in the agent's best interest to invest more effort into desirable versus undesirable features?*

We define $\beta$-desirability that measures the ratio of investment in features in $\mathcal{D}$ vs $\mathcal{U}$:

**Definition 1** ($\beta$-desirable effort profiles)**.** *Given $0 < \beta \leq 1$, an exogenous effort profile $e$ is said to be $\beta$-"desirable" if and only if: $\|e_{\mathcal{D}}\|_2 \geq \beta\|e\|_2$, i.e., the magnitude of effort towards desirable features is at least a $\beta$-fraction of the total effort.*

From the principal's point of view, incentivizing $\beta$-desirable effort profiles is not straightforward since agents are strategic, and may prefer undesirable features if they are low-cost to manipulate.

## 3  The Complete Information Setting

In the complete information setting, the agent knows precisely the true classifier $h_0$ deployed by the principal—equivalently, her prior $\Pi_h$ satisfies $\bar{h} = h_0$ (the mean belief matches the true classifier) and the covariance is given by $\Sigma_h = \mathbf{0}$ (there is no uncertainty). She also fully knows the causal graph $\mathcal{G}$—i.e., $\bar{w} = w$ and $\Sigma_w = \mathbf{0}$. Therefore, in the complete information case, the deterministic tuple $(h_0, \mathbb{C})$ is enough to fully characterize agent beliefs.

When the agent has no uncertainty about either the classifier or the causal graph, the agent's optimization problem (2) can be written as:

$$e^\star(h_0, \mathbb{C}) = \arg\min_{e \geq 0} \quad \mathsf{Cost}(e) \quad \text{s.t.} \quad (\mathbb{C}h_0)^\top e \geq \alpha. \tag{3}$$

I.e., the agent must find the minimum-cost effort profile that passes the (known) classifier $h_0$. Note that it suffices to only focus on non-negative efforts for all features (this is without loss of generality as we prove in Proposition 1). Program (3) is a convex optimization problem because the objective is convex for our cost functions and all constraints are linear. As such, this program can be solved efficiently. All proofs for this section can be found in Appendix C.

**Characterization of Optimal Effort Profiles.**   We now present the first main result (Theorem 1) where we characterize the structural properties of the optimal effort profile $e^\star$ for the cost function class outlined in Eq. (1) for $\alpha > 0$ (the case where $\alpha \leq 0$ is trivial, as we show in Proposition 2). Importantly, we highlight how the structure of the effort profile changes fundamentally when the cost function transitions from the $p = 1$ to the $p > 1$ regime.

**Theorem 1.** *The optimal effort profile $e^\star$ of an agent with weighted $\ell_p$-norm costs ($p \geq 1$) has the following structure:*

*(a) when $p = 1$, there always exists an optimal effort profile in which the agent needs to modify exactly one feature to pass the classifier $h_0$. The optimal feature to modify $f^*$ is the one which offers the best ratio of contribution to cost, i.e., $f^* \in \arg\max_{f \in \mathcal{F}} \frac{(\mathbb{C}h_0)_f}{c_f}$, with the magnitude of effort invested into feature $f^*$ given by:*

$$e^\star_{f^*} = \frac{\alpha}{(\mathbb{C}h_0)_{f^*}}.$$

*(b) Further, when $p > 1$, the optimal effort profile invests effort along all non-trivial features (with $(\mathbb{C}h_0)_f > 0$), with the magnitude of effort invested into feature $f$ satisfying:*

$$e_f^\star \propto \left( \frac{(\mathbb{C}h_0)_f}{c_f} \right)^{1/(p-1)} .$$

**Conditions for $\beta$-desirability.** In this segment, we derive necessary and sufficient conditions under which rational strategic agents have natural incentives to invest only in $\beta$-desirable effort profiles (profiles where a significant amount of effort is exerted on *desirable* features).

**Theorem 2.** *For any $\beta \in (0, 1]$ and a $\ell_p$-norm cost function (where $p \geq 1$), the agent's best response is always a $\beta$-desirable effort profile if and only if:*

*(a)* $\max_{f \in \mathcal{U}} \frac{(\mathbb{C}h_0)_f}{c_f} < \max_{f \in \mathcal{D}} \frac{(\mathbb{C}h_0)_f}{c_f}$ *when $p = 1$; and*

*(b)* $\left[ \sum_{f \in \mathcal{D}} \left( \frac{(\mathbb{C}h_0)_f}{c_f} \right)^{2/(p-1)} \right]^{1/2} \geq \frac{\beta}{\sqrt{1-\beta^2}} \left[ \sum_{f \in \mathcal{U}} \left( \frac{(\mathbb{C}h_0)_f}{c_f} \right)^{2/(p-1)} \right]^{1/2}$ *when $p > 1$,*

*with $\mathcal{D}$ and $\mathcal{U}$ representing the set of desirable and undesirable features respectively.*

The above result finds that, provided the principal designs a classifier $h$ such that the overall importance of desirable features is large enough, desirable behavior can *always* be incentivized. It also underlines the dependence on the causal graph, which is captured through $\mathbb{C}$. The causal graph characteristics determine the extent to which desirable behavior can be incentivized in any particular setting. This helps us to generate key insights about the design space of *desirable classifiers* (i.e., classifiers which can induce $\beta$-desirable effort profiles).

**Desirable Classifiers and Where to Find Them.** So far, we have provided conditions using which given any classifier $h_0$, we can check whether it incentivizes desirable effort profiles from strategic agents. However, this does not answer the question of how difficult it is to find a *desirable classifier*. Our following result answers this question.

**Theorem 3.** *For any $p \geq 1$, there always exists an instance of the problem $(\mathbb{C}, h_0)$ and $\beta > 0$ such that the space of $\beta$-desirable classifiers $\mathcal{H}$ is a non-convex set. However, when $|\mathcal{D}| = 1$, $\mathcal{H}$ can be shown to be convex for any $\beta > 0$ and any $\ell_p$-norm cost function with $p \in [1, 3]$.*

The first part of the theorem tells us that searching for desirable classifiers that also optimize for accuracy is expected to be difficult. This is because optimizing over non-convex sets is generally known to be computationally hard. For a more detailed formal discussion on hardness, see Appendix C.5.

However, the second part of the theorem argues that under the special case where there is only one desirable feature, the space of desirable classifiers is convex, at least for a limited subclass of $\ell_p$-norm cost functions. Not only does this circumvent the technical difficulty encountered in the general case, it also has other practical benefits. Note that this special case is actually aligned with what we expect in real life: indeed, a principal may define for themselves which feature they want to incentivize, and focus on one feature where they would really like to see improvements, especially if this is a feature that has historically not been properly leveraged. Not only that, but by targeting a single feature, the principal lowers the agents' cognitive load for best-responding, which is always desirable in practice. Note that we do provide a convexification heuristic for cases where $|\mathcal{D}| > 1$ that tries to induce desirable effort by minimizing the importance of undesirable features (See Appendix C.4 for details).

## 4 Incomplete Information Setting

Recall that for the incomplete information setting, the agent's optimization problem is:

$$e^\star(\Pi_h, \Pi_{\mathbb{C}}) = arg \min_e \quad \mathsf{Cost}(e) \quad \text{s.t.} \tag{4}$$

$$\mathbb{P}_{h \sim \Pi_h, \mathbb{C} \sim \Pi_{\mathbb{C}}} \left[ (\mathbb{C}h)^\top e \geq \alpha \right] \geq 1 - \delta, \quad \delta \in (0, 1).$$

**Models of Information.** Recall that there can be two sources of uncertainty: i) the principal's classifier; and, ii) the edge weights of the causal graph $\mathcal{G}$ (the graph topology is assumed to be common knowledge). In particular, this leads to the following three incomplete information models:

(1) Uncertainty only exists in the principal's classifier, the causal graph is fully known;

(2) Uncertainty only exists in the edge weights of the causal graph, the classifier is fully known;

(3) Uncertainty exists over both the classifier and the causal graph.

We refer to models 1 and 2 as models of *partially incomplete information* while model 3 will be referred to as a model of *total incomplete information*. The proofs for this section are in Appendix D.

**Optimal Effort Computation.** We note that, unlike the complete information setting, the agent's optimization problem is significantly more involved. Therefore, our first objective here is to answer the following question: *Under what degree of incomplete information can agents still compute their best response efficiently ?* This is crucial because if agents cannot best respond reliably, the question of designing classifiers that incentivize desirable agent behavior is irrelevant.

**Theorem 4.** *For $\delta \leq 1/2$, the agent's optimization problem* (4) *is:*

*(a) always convex, when uncertainty only exists in the classifier (model 1);*

*(b) convex, when uncertainty only exists in the causal graph and the causal graph has a bipartite structure (i.e., special cases of model 2)*

*(c) non-convex, for all other cases of model 2 with general causal graph structures and all cases of model 3 with total uncertainty.*

*In scenarios (a) and (b) above, we have convexity because the overall uncertainty $\mathbb{C}h$ turns out to be multi-variate Gaussian, i.e., $\mathbb{C}h \sim \mathcal{N}(\mu_{\mathbb{C}h}, \Sigma_{\mathbb{C}h})$, in which case, the agent's optimization problem reduces to the following convex program:*

$$e^\star(\Pi_h, \Pi_{\mathbb{C}}) = \arg\min \quad \textbf{Cost}(e) \quad s.t. \quad \alpha - \mu_{\mathbb{C}h}^\top e - p_\delta \cdot \sqrt{e^\top \Sigma_{\mathbb{C}h} e} \leq 0, \tag{5}$$

*where $p_\delta = \Phi^{-1}(\delta)$ and $\Phi^{-1}(\cdot)$ is the inverse of the standard normal CDF.*

The above result highlights that under limited uncertainty, the agent can still efficiently solve for an effort profile that helps her to pass the classifier with high probability. Importantly, note that the above result is quite general: any setting where the agent's prior on the feature importance vector $\mathbb{C}h$ is Gaussian is captured by our framework. In particular, we allow for largely different models of incomplete information from the ones we have defined so far: for example, the principal may choose to reveal some information about the importance of features to agents, or agents may form priors about the feature importance vector directly through interactions with peers[7].

We also want to highlight that unlike the complete information case where the agent's optimization problem is always feasible (i.e., the agent can always pass the classifier by choosing effort correctly), *under uncertainty, a positive outcome is not guaranteed.* We provide a complete characterization of the feasibility of Problem (5) in the Appendix (Proposition 6). The intuition is that there is a trade-off between the degree of uncertainty for the agent and the maximum coverage probability $(1 - \delta)$ that can be achieved under that uncertainty.

**Characterization of Optimal Effort Profiles.** We have already identified settings with partial uncertainty where the agent's optimization problem is convex and tractable. While it is much harder to compute best responses in closed form because of the involved nature of the optimization problem, we still provide insights about structural properties of the optimal effort profile. We identify key differences with the complete information setting, for example, under partial uncertainty, the optimal effort profile for agents with weighted $\ell_1$ costs may always involve investment in more than one feature (Proposition 7). In the following result, we focus specifically on the $\ell_2$-cost case where we can actually characterize the best response in semi-closed form:

**Theorem 5.** *For $\ell_2$-norm cost functions, the effort profile $e^\star$ which is the optimal solution to Problem (5), is of the following form:*

$$e^\star = \lambda^* (k_1 I + k_2 \Sigma_{\mathbb{C}h})^{-1} \mu_{\mathbb{C}h},$$

---

[7]One could argue that agents forming priors directly on the feature importance vector is more reasonable in practice because it foregoes the need for agents to reason about how the causal graph and classifier interact, thereby reducing the cognitive load required to arrive at an optimal decision.

where $k_1, k_2, \lambda^* > 0$. *Further, if $\Sigma_{\mathbb{C}h}$ is a diagonal matrix with entry $(\Sigma_{\mathbb{C}h})_f$ corresponding to feature $f$, then*

$$e_f^\star = \frac{\lambda^*(\mu_{\mathbb{C}h})_f}{k_1 + k_2 \cdot (\Sigma_{\mathbb{C}h})_f} \quad \forall\, f \in \mathcal{F}.$$

The special case of $\Sigma_{\mathbb{C}h}$ being diagonal offers good intuition into how agents with $\ell_2$ costs would exert effort under partial uncertainty. It shows that the agent is expected to invest more effort into features with higher expected contribution $(\mu_{\mathbb{C}h})_f$. Further, the denominator highlights that agents may shy from features they have a lot of uncertainty about, thereby leading to lower effort into them.

We also note that diagonal $\Sigma_{\mathbb{C}h}$ arises in very natural settings. One such setting is scenario (b) in Theorem 4 when the uncertainty is only on the causal graph $\mathcal{G}$ and the causal graph is bipartite, i.e., features are either *causal* (they affect other features, but cannot be affected themselves) or *proxy* (they are affected by causal features, but cannot affect any other feature). In this case, causal features only have outgoing edges, while proxy features only have incoming edges. We prove this formally in Proposition 8 in Appendix D. Bipartite causal graphs are standard assumptions in much of the strategic classification literature [1, 26].

**$\beta$-desirability under $\ell_2$-costs with Incomplete Information.** We conclude this section with a discussion on how to induce $\beta$-desirable effort profiles under incomplete information. As we see so far, it may be difficult to characterize the agent's optimal effort in closed form under incomplete information, except for some limited cases. We focus on providing broad insights here, and build on this discussion through numerical experiments in Section 5.

*The Interpretable Case of Diagonal Covariance $\Sigma_{\mathbb{C}h}$:* In this special setting, we can identify conditions that guarantee investment in $\beta$-desirable effort profiles by rational agents. We present the following two results:

**Corollary 1.** *Suppose that $\Sigma_{\mathbb{C}h}$ is a diagonal matrix. In that setting, if all features have the same overall level of uncertainty and the mean feature importance vector $\mu_{\mathbb{C}h}$ satisfies:*

$$\| (\mu_{\mathbb{C}h})_{\mathcal{D}} \|_2 \geq \frac{\beta}{\sqrt{1 - \beta^2}} \| (\mu_{\mathbb{C}h})_{\mathcal{U}} \|_2,$$

*then the best response of a rational agent with $\ell_2$-norm cost is to invest in a $\beta$-desirable effort profile.*

**Corollary 2.** $e_f^\star = \frac{\lambda^*(\mu_{\mathbb{C}h})_f}{k_1 + k_2 \cdot (\Sigma_{\mathbb{C}h})_f}$ *is decreasing in $(\Sigma_{\mathbb{C}h})_f$, therefore lower levels of uncertainty in desirable features favors effort profiles with a higher degree of desirability $\beta$.*

The first corollary follows directly from Theorem 5 and should be intuitive—when agents face the same degree of uncertainty about all features, they choose which features to invest effort in based on the mean importance of the features. Therefore, it makes sense that higher the total net importance (measured by the $\ell_2$-norm) of the set of desirable features, higher the incentive for agents to invest in desirable effort profiles. On the other hand, when different features have different levels of uncertainty, less uncertainty on desirable features is good for $\beta$-desirability. This is because having a higher degree of uncertainty (higher variance $(\Sigma_{\mathbb{C}h})_f$) about the importance of a feature actively discourages agents from investing effort into said feature.

*Non-diagonal Covariance $\Sigma_{\mathbb{C}h}$:* We explore the non-diagonal $\Sigma_{\mathbb{C}h}$ case in Section 5, with experiments on real data that consider more general cases of $\Sigma_{\mathbb{C}h}$ not being diagonal, specifically covering Model 1, when the classifier is not fully known to an agent. Our experiments suggest that many of the same insights about $\beta$-desirability hold (even *without* the assumption that $\Sigma_{\mathbb{C}h}$ is diagonal).

## 5 Numerical Experiments

Our experimental study focuses on a setting where the learner is trying to reduce a population's risk of cardiovascular disease. To do so, we identify relevant features and build a causal graph based on the recent medical study of [19]. Their study aims to identify the causal links between features such as smoking, diet, or obesity, and whether a patient may develop a cardiovascular disease (CVD). We exclude immutable features such as age and ethnicity, focusing instead on eight modifiable features: alcohol consumption, diet, physical activity, smoking, diabetes mellitus (DM), hyperlipidemia (HPL), hypertension (HPT), and obesity. Among these, we designated as desirable

the features corresponding to lifestyle interventions—namely, alcohol, diet, physical activity, and smoking—over those corresponding to medical conditions or interventions (DM, HPL, HPT, and obesity). The full experimental setup is detailed in Appendix B.1.

**Deployed classifiers:** We consider four[8] mean beliefs $\mu_h$ on the vector $h$, that we denote as:

- *DM*: There is a weight of $1$ on the "DM" feature, and $0$ on all others.
- *HPL*: There is a weight of $1$ on the "HPL" feature, and $0$ on all others.
- *HPT*: There is a weight of $1$ on the "HPT" feature, and $0$ on all others.
- *Obesity*: There is a weight of $1$ on the "Obesity" feature, and $0$ on all others.

For each of the four classifiers described above, we document the *mean contribution* of each feature, given by $\mu_{\mathbb{C}h} = \mathbb{C}\mu_h$ and the $\ell_2$ norm of the mean contribution over the set of desirable and undesirable features, given by $\ell_2(\mathcal{D})$ and $\ell_2(\mathcal{U})$ respectively. All values are recorded in Table 1.

| Classifier | Alcohol | Diet | Activity | Smoking | DM | HPL | HPT | Obesity | $\ell_2(\mathcal{D})$ | $\ell_2(\mathcal{U})$ |
|---|---|---|---|---|---|---|---|---|---|---|
| DM | 0.1 | 0.84 | 0.82 | 0.52 | 1 | 0 | 0 | 0 | 1.28 | 1 |
| HPL | 0.14 | 0.84 | 0.82 | 0.34 | 0 | 1 | 0 | 0 | 1.23 | 1 |
| HPT | 0.62 | 0.84 | 0.82 | 0.86 | 0 | 0 | 1 | 0 | 1.58 | 1 |
| Obesity | 0.64 | 0.86 | 0.82 | 0 | 0 | 0 | 0 | 1 | 1.35 | 1 |

Table 1: Mean contribution vector $\mu_{\mathbb{C}h}$ for the 4 classifiers: DM, HPL, HPT, Obesity

We now make some key observations:

**Desirable features can be incentivized even if they are never observed.** Figures 2 and 3 demonstrate that agents choose to invest significant effort into desirable features even if in our four classifiers of choice, no weight has been put on any of the features in set $\mathcal{D}$ (see Appendix B.4 for details).

**Effect of total contribution on desirable vs undesirable features:** All four classifiers incentivize greater effort on the set of desirable features $\mathcal{D}$ compared to the set of undesirable features $\mathcal{U}$ (see Table 1). In Figure 2, we observe that all classifiers achieve $\beta > 0.5$ under low to moderate uncertainty, which aligns with the prediction of Corollary 2. Furthermore, we find that hypertension (HPT) outperforms Obesity, which outperforms diabetes mellitus (DM), which then outperforms hyperlipidemia (HPL), in terms of effectiveness at incentivizing effort on desirable features. This ranking is consistent with the ordering of $\ell_2(\mathcal{D})$ values across the classifiers and reinforces the theoretical insights provided by the special diagonal covariance case outlined in Theorem 5.

**Effect of uncertainty level $\sigma$.** In Figure 2, we plot how $\beta$ varies as a function of the uncertainty parameter $\sigma$ for different classifiers. Higher $\sigma$ indicates a higher degree of uncertainty about the classifier. As $\sigma$ increases, all four classifiers degrade in terms of desirability ($\beta$). This is intuitive: at higher levels of uncertainty, the contribution of desirable features sees higher variance, as they affect not only themselves but also other features. Undesirable features then become safer to modify.

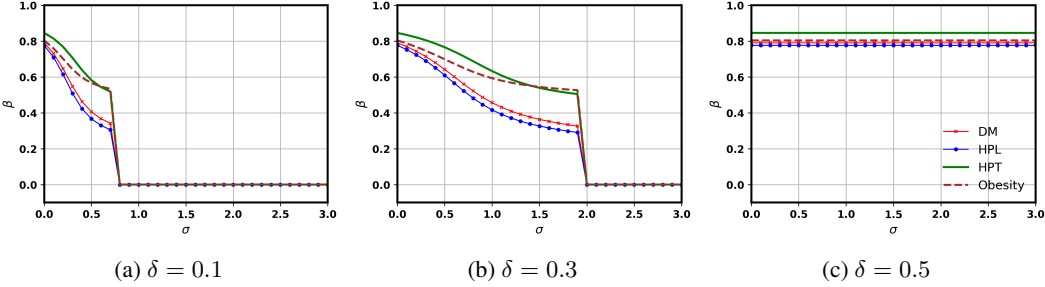

(a) $\delta = 0.1$        (b) $\delta = 0.3$        (c) $\delta = 0.5$

Figure 2: Plot of how $\beta$ varies with $\sigma$ at different levels of $\delta$ and for different classifiers.

**Effect of the failure probability $\delta$.** At a fixed level of uncertainty $\sigma$, as the failure probability $\delta$ increases, $\beta$ improves across all four classifiers (Figure 3). This is expected—higher $\delta$ means that the

---

[8]We note that all other beliefs that only use undesirable, observable features are a linear combination of the four beliefs above, so our insights extend to general classifiers.

agent is less stringent on the coverage probability requirement and hence has a much larger space of feasible effort profiles to choose from—including desirable ones with a high net contribution.

**Trade-offs between $\sigma$ and $\delta$.** The failure probability $\delta$ is closely related to the level of uncertainty $\sigma$. At a fixed level of uncertainty $\sigma$, there is a limit on how low a failure probability $\delta$ can be achieved (Figure 3). Similarly, in order to achieve a given failure rate $\delta$, there is a maximum amount of uncertainty $\sigma$ that can be tolerated (Figure 2); beyond that the problem becomes infeasible. This tracks our theoretical findings in Proposition 6. At $\delta = 0.5$, uncertainty becomes irrelevant since the agent only needs a $50\%$ chance of passing the classifier and can rely solely on the mean belief $\mu_h$.

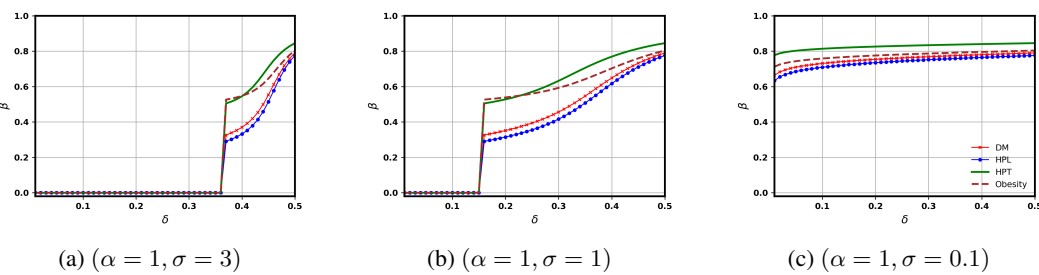

(a) $(\alpha = 1, \sigma = 3)$       (b) $(\alpha = 1, \sigma = 1)$       (c) $(\alpha = 1, \sigma = 0.1)$

Figure 3: Plots of how $\beta$ varies with $\delta$ for different parameter combinations.

**Effect of $\alpha$.** $\alpha$ represents the amount by which the agent is shy of a positive classification outcome. In this case, $\beta$ has no dependence on $\alpha$ (see Figure 5 in Appendix B.3). This is an artifact of the $\ell_2$-norm cost function — the agent's optimal effort profile is proportional to $\alpha$ in each feature and therefore $\beta$ remains unaffected. However, note that $\alpha$ does affect the cost incurred by the agent, the farther she is from the classification boundary (higher $\alpha$), the higher is the cost incurred.

## 6 Discussion

In this paper, we adopt a causal perspective to the problem of strategic classification. The principal deploys a linear classifier which classifies agents "positive" or "negative" based on a set of features embedded on a *causal graph*. Since agents are strategic, they are expected to invest effort "cleverly" to modify their features in the hopes of a positive classification outcome while incurring the minimum possible cost. Therefore, understanding how agents respond when they have different levels of access to information about the deployed classifier and the causal graph, is significant to the principal from the perspective of classifier design. We try to answer this central question from the principal's point of view: *how to design a classifier which incentivizes agents to invest in desirable effort profiles?*

There are many relevant avenues for future work. Our model of uncertainty involves agents with Gaussian priors over the classifier or the edge weights of the causal graph or both, under the assumption that the graph topology is fully known. In reality, there may be other forms of uncertainty—for example, given a large number of features, it may be unreasonable to assume that agents have complete knowledge about all causal relationships between features. It may be interesting to explore other, more interpretable information structures—e.g., instead of independent priors on the classifier and the causal graph, agents may have access to a partial ordering on features by relative importance. Such models may be closer to how humans perceive and respond to uncertainty in reality. Finally, different populations with different levels of uncertainty in their priors may have markedly different abilities to respond to the classifier. The fairness implications of such information asymmetries are an important direction for future work.

## Acknowledgements

The authors are grateful from support from the US National Science Foundation (NSF) under grants IIS-2504990, IIS-2336236, and an Amazon Research Award. This work has also been partially supported by project MIS 5154714 of the National Recovery and Resilience Plan Greece 2.0 funded by the European Union under the NextGenerationEU Program. Any opinions and findings expressed in this material are those of the authors and do not reflect the views of their funding agencies.

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

# Supplementary Material for paper
## *"Incentivizing Desirable Effort Profiles in Strategic Classification: The Role of Causality and Uncertainty"*

## A  Related Work

**Strategic classification.**  Strategic classification has been widely studied in recent years. This belongs to a broad class of problems in economics called principal-agent problems where agents act strategically in their self-interests which are often misaligned with the principal's interests [16, 29, 38, 39]. Early works in strategic classification focused on scenarios where agents manipulate their observable features to "game" a published classifier, thereby increasing their chances of a favorable label without genuinely improving underlying attributes [1, 8, 9, 11, 17, 30, 41, 44]. [5, 6, 18, 26, 40] move away from this assumption and consider settings where agents can not only game the classifier, but also invest in real improvements. In many cases, actual improvement involves investing effort which is not directly observable by the principal — this again has similarities to the notion of moral hazard in insurance markets [3, 35] and other general settings [4]. Fairness in strategic classification [14, 21, 33]) has also been an important avenue of work, but is less related to this work.

**Causality.**  A useful tool to model manipulations as opposed to effective improvement is *causality* [22, 36]. In strategic classification, a few recent studies have incorporated causal modeling to account for interdependencies among features [2, 5, 20, 26, 32, 40]. [32] highlights that in strategic classification, learning a classifier that incentivizes gaming as opposed to improvement is as hard as learning the underlying causal graph between features. [40] and [5] both focus on special cases of linear causal graphs, unlike our work that considers general cases of linear graphs. [2] explore strategic classification using a different structural framework known as "manipulation graphs," where each agent has a fixed set of costly effort profiles that they may choose from, also defining a special case of causal graphs. [20] distinguish among causal, non-causal, and unobserved features, but focus on a different objective: predictive accuracy.

Perhaps closest to us is the work of [26]. Like us, they focus on general causal graphs; however, we highlight several major differences. First, we focus on *classification* settings, while [26] focus on regression and scoring settings. Second, we highlight differences in our agent model, where our agents invest effort to pass the classifier with reasonably high probability, while agents in [26] *always* exert effort to improve their score. Third, we note that our cost model is strictly more general: where [26] focuses on linear costs, our work considers general $\ell_p$-costs[9]. Finally, unlike [26], our study incorporates *incomplete information*, where agents may not fully understand either the causal graph or the deployed classifier.

**Incomplete Information.**  A closely related line of work investigates strategic classification under varying models of *information* available to agents. In many real-world settings, agents may have *incomplete information* about the classifier—either because it is too complex, or because the learner's model is proprietary, or the causal relationships governing feature interactions [6, 10, 15]. Or, agents might misperceive the classifier due to behavioral biases [12]. However, to the best of our knowledge, we are the first work in the space of strategic classification to incorporate *both* causal modeling and incomplete information in strategic classification.

**Algorithmic recourse.**  Our work is also closely related to the extensive literature on algorithmic recourse [24, 25, 28, 42, 43]. From a mathematical standpoint, strategic classification and algorithmic recourse can be seen as flip sides of each other: in recourse, the learner tells the agent what actions to take (but the agents may decide to take a different action unless properly incentivized) while in strategic classification, the agents decide on their own actions based on the classifier. We note, however, several differences between recourse and strategic classification: a) From a practical point of view, recourse requires providing a recommended action to each agent who obtains a negative

---

[9]Our results show that this choice of cost is important, noting a sharp distinction in agent behavior between the cases of $\ell_1$-cost and $\ell_p$ costs for $p > 1$.

decision, and is still less common in practice than releasing a single, often not fully transparent, model. Strategic classification, especially with incomplete information, allows us to capture such practical scenarios. b) The addition of the $\beta$-desirability constraint is novel. As we see in many places throughout the paper, this constraint makes the problem significantly more difficult, leading to non-convexities. We expect these issues to also arise when adding desirability to the algorithmic recourse literature, since the optimization problems solved by both fields are similar, and we think that the inclusion of desirability in the recourse literature is an exciting direction for future work. c) To the above point, much of the recourse literature aims to find a low-cost path between an agent's current features and features that lead to positive classification. Importantly, much of the literature does not think specifically about whether that path involves gaming or modifying undesirable features. [28] in fact show that standard counterfactual recourse algorithms often lead to undesirable outcomes such as gaming the system. This is an important limitation: as agents game the classifier, the classifier loses in accuracy and robustness, and must be modified to compensate for this accuracy loss, reducing the effectiveness of the recourse. This includes *causal* algorithmic recourse, such as [24], where the causality is typically used to ensure accuracy of the recourse by taking into account how different features affect each other, whereas previous work that effectively assumed independence of features and could mis-estimate how proposed recourse would translate into feature changes. In much of this literature, however, causality is not introduced to prevent gaming vs improvement or understand desirability.

# B  Additional Details for Experimental Section 5

## B.1  Experimental setup

**Identifying Relevant Features**   We identify a subset of 8 features used in [19] that we focus on in our experimental study. In particular, we did not include features that cannot be changed such as age or ethnicity, and only include the features that can be modified by an individual. The 8 features we identified are: alcohol consumption, diet, physical activity, smoking, diabetes mellitus (DM), hyperlypidia (HPL), hypertension (HPT), and obesity. We normalized features to be between $[0, 1]$[10].

Among these features, and as noted in our introduction, a principal (i.e., a doctor) would like to incentivize people to focus on preventative, lifestyle interventions over medical treatment interventions. Hence, we separate them to desirable and undesirable to modify as follows:

- *Desirable:* Alcohol, Diet, Activity, Smoking. Note that desirable means here that these features are desirable to *modify*, not that, for example, alcohol consumption is desirable.

- *Undesirable:* DM, HPL, HPT, Obesity. Note that these features are not "undesirable" per se, but rather less desirable than lifestyle interventions.

**Building the Causal Graph:**   The study of [19] asked the experts to report the likelihood of causation on a Likert scale from 1 to 7, which is then transformed into "fuzzy score" via the Fuzzy Delphi Method [31]. We denote this score $s$. A fuzzy score of $0.5$ and above indicates that experts at least moderately agree with a relationship being causal, with an increasing score $s$ indicating stronger agreement. A fuzzy score of $0.5$ or below indicates that the experts at best disagree with the feature being causal, with the strength of the disagreement increasing as the score goes down.

We follow the expert agreement of [19] to build our causal links. Specifically, we identify a link as causal if and only if $s > 0.5$. Further, since a score of $0.5$ denotes that experts are at the boundary of agreeing vs disagreeing on causality, we renormalize our scores to be between $0$ and $1$: to do so, we apply a linear transformation that maps $s = 0.5$ to a causal weight of $0$, and $s = 1$ to a causal weight of $1$. We obtain the following graph (Figure 4):

*Generating Prior Beliefs* We note that our desirable features are generally harder to observe than our undesirable features. First, DM, HPL, HPT, and Obesity are easy-to-quantify features that are also verifiable by a doctor (e.g., though blood work). On the other hand, lifestyle habits are not only hard to observe, but also often mis-reported to clinicians (i.e., under-reporting alcohol consumption, or

---

[10]For simplicity and wlog, we assume that 0 is the least "healthy" value of the feature, and 1 is the "healthiest" value of the feature. For example, for smoking, 1 maps to not smoking; for activity, 1 maps to high amount of weekly physical activity.

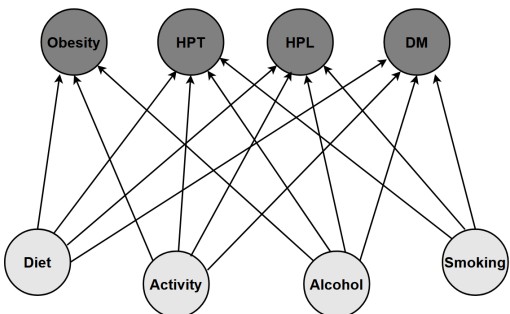

Figure 4: Causal graph of features which affect the output of interest "Risk of Cardio-vascular disease (CVD)". There are $8$ features, all of which are *causal*. The features at the bottom form the set of desirable features $\mathcal{D}$ and those on the top form the set of undesirable features $\mathcal{U}$. The causal links are indicated on the graph. This causal graph has a special structure: it is *bipartite*. The edge weights are recorded in Table 2 in Appendix B.

lying about smoking to avoid insurance upcharges). Hence, we generate all our priors of $h$ to have both a mean and variance of $0$ for all desirable features (i.e., it is fully known that desirable features are not observed by a clinician, and so not used in the clinician's classifier for high risk of CVD), as they are effectively *unobservable*. We consider four mean beliefs $\mu_h$ on the vector $h$, that we denote as follows:

- *DM*: There is a weight of $1$ on the "DM" feature, and $0$ on all others.

- *HPL*: There is a weight of $1$ on the "HPL" feature, and $0$ on all others.

- *HPT*: There is a weight of $1$ on the "HPT" feature, and $0$ on all others.

- *Obesity*: There is a weight of $1$ on the "Obesity" feature, and $0$ on all others.

We note that all other beliefs that only use undesirable, observable features are a linear combination of the four beliefs above, so our insights extend to general classifiers. Also, we demonstrate later that while the classifier does not put any weight on unobserved/desirable features, agents may still exert effort on them because they affect the observed/undesirable features used in the classifier.

The variance of each of the desirable features is taken to be $0$ (there is complete information that no weight is put on these features in the principal's classifier). Further, we assume that all undesirable features have the same variance, which is parametrized by $\sigma > 0$ — thus $\sigma$ is a measure of the level of incomplete information. Finally, the covariance matrix of the classifier ($\Sigma_h$) is taken to be diagonal for simplicity of exposition and interpretation, i.e., individuals' beliefs do not encode correlations between features in the deployed classifier.

## B.2  Supplementary Tables

| Features | Alcohol | Diet | Activity | Smoking | DM | HPL | HPT | Obesity |
|----------|---------|------|----------|---------|------|------|------|---------|
| Alcohol | 0 | 0 | 0 | 0 | 0.10 | 0.14 | 0.62 | 0.64 |
| Diet | 0 | 0 | 0 | 0 | 0.84 | 0.84 | 0.84 | 0.86 |
| Activity | 0 | 0 | 0 | 0 | 0.82 | 0.82 | 0.82 | 0.82 |
| Smoking | 0 | 0 | 0 | 0 | 0.52 | 0.34 | 0.86 | 0 |
| DM | 0 | 0 | 0 | 0 | 0 | 0 | 0 | 0 |
| HPL | 0 | 0 | 0 | 0 | 0 | 0 | 0 | 0 |
| HPT | 0 | 0 | 0 | 0 | 0 | 0 | 0 | 0 |
| Obesity | 0 | 0 | 0 | 0 | 0 | 0 | 0 | 0 |

Table 2: Adjacency matrix $A$ which captures the edge weights of the causal graph in Figure 4

## B.3 Effect of $\alpha$ on $\beta$.

As explained earlier, $\beta$ is independent of $\alpha$. We demonstrate this below by plotting $\beta$'s as a function of $\delta$ for two different values of $\alpha$: $\alpha = 1$ and $\alpha = 10$.

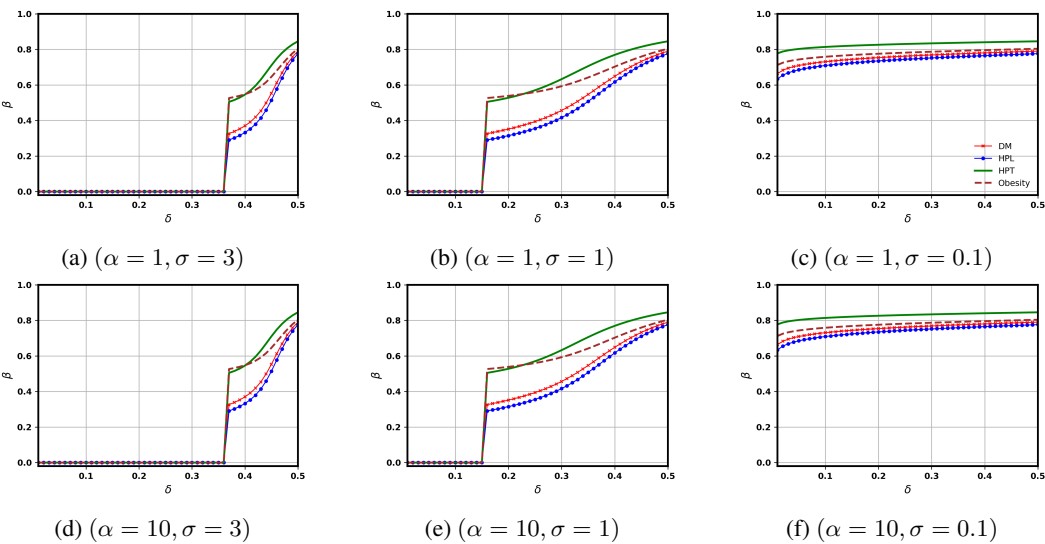

(a) $(\alpha = 1, \sigma = 3)$    (b) $(\alpha = 1, \sigma = 1)$    (c) $(\alpha = 1, \sigma = 0.1)$

(d) $(\alpha = 10, \sigma = 3)$    (e) $(\alpha = 10, \sigma = 1)$    (f) $(\alpha = 10, \sigma = 0.1)$

Figure 5: Plots of how $\beta$ is invariant with $\alpha$ at all levels of uncertainty.

## B.4 Further discussion on results

**Desirable features can be incentivized even if they are never observed.** In our four classifiers of choice, observe that no weight has been put on any of the features in set $\mathcal{D}$ because they represent features which cannot be directly observed. However, Figures 2 and 3 demonstrate that agents still choose to invest significant effort into desirable features. This is a direct result of the *causal relationship between features*. Observe that in the causal graph, the desirable features influence multiple undesirable features simultaneously. This means that an agent obtains a larger improvement in the undesirable features (which actually affect the agent's classification), not by modifying them directly, but by investing effort into desirable features.

## C  Results & Proofs for the Complete Information Setting

**Proposition 1.** *For the cost functions defined in* (1)*, **without loss of generality**, we can assume that:* $(\mathbb{C}h_0)_f \geq 0$ *and that* $(e)_f \geq 0 \ \forall f \in \mathcal{F}$.

*Proof.* Let $e^\star$ be the optimal effort profile for the agent, i.e., the profile that corresponds to the solution of (3). First, note that for any feature $f \in \mathcal{F}$, $(\mathbb{C}h_0)_f = 0$ implies $e_f^\star = 0$. Indeed, if feature $f$ has no contribution towards the classification decision, then no effort should be expended on $f$ in the optimal effort profile.

Now, we can focus on features for which $(\mathbb{C}h_0)_f \neq 0$. When $(\mathbb{C}h_0)_f < 0$, we will show that $e_f^\star \leq 0$. Suppose that $e_f^\star > 0$. In this case, we can construct a new effort profile $e'$ as follows: $e'_f = 0$ and $e'_g = e_g^\star$ for all $g \in \mathcal{F}, g \neq f$. It is easy to see that $e'$ is still feasible but $\mathsf{Cost}(e') < \mathsf{Cost}(e^\star)$ which contradicts the fact that $e^\star$ is the optimal solution. Therefore, $e_f^\star \leq 0$. Similarly, we can show that when $(\mathbb{C}h_0)_f > 0$, $e_f^\star \geq 0$.

The above discussion implies that whenever $(\mathbb{C}h_0)_f \neq 0$, then $(\mathbb{C}h_0)_f e_f^\star \geq 0$ - therefore, without loss of generality, it suffices to assume $(\mathbb{C}h_0)_f > 0$ and only search over the space $e \geq 0$ (because the optimal solution $e^\star \geq 0$). This concludes the proof. $\qquad \square$

**Proposition 2.** *When $\alpha \leq 0$, then $e^\star = 0$ which means that the agent does not need to invest any effort to change her features.*

*Proof.* Note that the objective value is greater than or equal to zero since $c_f > 0$ for all $f \in \mathcal{F}$ and $e \geq 0$. But, since $\alpha \leq 0$, $e = 0$ is feasible to the above problem and it also achieves an objective value of $0$. Therefore, $e = 0$ must be optimal. $\square$

## C.1    Proof of Theorem 1

We complete the proof in two parts: first, we prove the $p = 1$ case in Lemma 1 followed by the $p > 1$ case in Lemma 2.

**Lemma 1.** *When $\alpha > 0$, there exists an optimal effort profile for the agent in which she needs to modify **exactly one feature** to pass the classifier $h_0$. The optimal feature to modify $f^*$ is the one which offers the best ratio of contribution to cost, i.e.,*

$$f^\star \in \arg\max_{f \in \mathcal{F}} \frac{(\mathbb{C}h_0)_f}{c_f},$$

*and the optimal amount of effort to be invested into that feature is given by:*

$$e_{f^\star} = \frac{\alpha}{(\mathbb{C}h_0)_{f^\star}}.$$

*Proof.* For $p = 1$, we have the following optimization problem for the agent:

$$\min_{e \geq 0} \quad c^\top e \quad \text{s.t.} \quad (\mathbb{C}h_0)^\top e \geq \alpha. \tag{6}$$

The optimization problem in (6) (which we will call the primal problem $P$) is a linear program whose feasible region is given by the following polyhedron $Q = \left\{ e \in \mathbb{R}^{n+k} : (\mathbb{C}h_0)^\top e \geq \alpha, e \geq 0 \right\}$. Our first goal is to argue that the optimal solution is a corner point of $Q$ which requires us to prove the following: i) firstly, $Q$ has at least one corner point, and ii) the optimal solution is bounded which would imply that it must be at a corner point of $Q$. Note that the polyhedron $Q$ has no line (because it is a subset of the positive orthant) and therefore, it must have at least one corner point[11]. We can now write the dual problem $(D)$ as follows:

$$\max_{\pi} \quad \alpha\pi \quad s.t. \quad (D)$$
$$\pi \cdot (\mathbb{C}h_0) \leq c \quad (e)$$
$$\pi \geq 0.$$

We know that the dual problem $(D)$ is feasible ($\pi = 0$ is feasible to $D$) which implies that the optimal solution to $(P)$ cannot be unbounded. Hence, we conclude that there must exist a corner point optimal solution to problem $(P)$. Now, note that all corner points of $Q$ are obtained by the intersection of the hyperplane $(\mathbb{C}h_0)^\top e \geq \alpha$ with the positive axes. So any corner point of $Q$ must be of the form where exactly one entry corresponding to some feature $f$ is positive (i.e., takes value $\frac{\alpha}{(\mathbb{C}h_0)_f}$) and all other entries are zero. This implies that there exists an optimal effort profile where the agent needs to modify exactly one feature, proving the first part of the lemma.

For proving the second part, we will use the complementary slackness conditions on the dual constraints. We already know that there exists an optimal primal solution where there is some feature $f^\star$ with $e_{f^\star} = \frac{\alpha}{(\mathbb{C}h_0)_{f^\star}}$ and $e_{f'} = 0$ for all $f' \neq f^\star$. Let $\pi^\star$ be the optimal dual solution. Using complementary slackness, we know that $\pi^\star \cdot (\mathbb{C}h_0)_{f^\star} = c_{f^\star}$ which implies that:

$$\pi^\star = \frac{c_{f^\star}}{(\mathbb{C}h_0)_{f^\star}}.$$

Since $\pi^\star$ must also be feasible to $(D)$, we must have:

$$\pi^\star \leq \frac{c_{f'}}{(\mathbb{C}h_0)_{f'}} \quad \forall f' \neq f^\star, (\mathbb{C}h_0)_{f'} \neq 0,$$

which implies that:

$$f^\star \in \arg\max_{f \in \mathcal{F}} \frac{(\mathbb{C}h_0)_f}{c_f}.$$

This concludes the proof of the lemma. $\square$

---

[11]For more details on the polyhedral theory related to linear optimization, please see [7].

**Lemma 2.** *When the cost function is the weighted $\ell_p$-norm of the effort for $p > 1$, the optimal effort profile for the agent $e^\star$ satisfies:*

$$e_f^\star \propto \left( \frac{(\mathbb{C}h_0)_f}{c_f} \right)^{1/(p-1)} \quad \forall\, f \in \mathcal{F}.$$

*Proof.* We solve the constrained optimization problem using Lagrange multipliers. Define the Lagrangian as follows:

$$\mathcal{L}(e, \pi) = \left( \sum_{f \in \mathcal{F}} c_f (e_f)^p \right)^{1/p} + \pi \left( -(\mathbb{C}h_0)^\top e + \alpha \right),$$

where $\pi$ is the Lagrange multiplier associated with the constraint as defined earlier. This gives us the following set of KKT conditions:

$$\nabla_e \mathcal{L}(e, \pi) = 0,$$
$$\pi \cdot \left( -(\mathbb{C}h_0)^\top e + \alpha \right) = 0,$$
$$\pi \geq 0,$$
$$\alpha - (\mathbb{C}h_0)^\top e \leq 0, \ e \geq 0.$$

Since our optimization problem is convex, it suffices to find a pair $(e^\star, \pi^\star)$ that satisfies the KKT conditions and we can automatically conclude that $e^\star$ is optimal to the primal problem.

First, we show that the constraint $(\mathbb{C}h_0)^\top e \geq \alpha$ must be active at the optimal solution. We prove this by contradiction. Suppose, if possible that $-(\mathbb{C}h_0)^\top e^\star + \alpha < 0$. However, this means that we can obtain the optimal solution $e^\star$ by solving the primal problem as if it were unconstrained. In that case, it must be that $e^\star = 0$, but observe that $e = 0$ is not even feasible (and hence cannot be optimal). This implies that the constraint must hold at equality. Therefore, we can solve for $e^\star$ and $\pi^*$ by solving the following system:

$$-(\mathbb{C}h_0)^\top e + \alpha = 0,$$
$$\nabla_e \mathcal{L}(e, \pi) = 0.$$

Now,

$$(\nabla_e \mathcal{L}(e, \pi))_f = \frac{\partial \mathcal{L}}{\partial e_f} = \frac{c_f (e_f)^{p-1}}{\left[ \left( \sum\limits_{f \in \mathcal{F}} c_f (e_f)^p \right)^{1/p} \right]^{p-1}} - \pi \cdot (\mathbb{C}h_0)_f.$$

We have already argued that $e^\star \neq 0$. Therefore, $\pi^\star > 0$. This implies that for all features $f \in \mathcal{F}$, whenever $(\mathbb{C}h_0)_f > 0$, we must have:

$$e_f^\star \propto \left( \frac{(\mathbb{C}h_0)_f}{c_f} \right)^{1/(p-1)},$$

and when $(\mathbb{C}h_0)_f = 0$, the condition holds trivially. This concludes the proof of the lemma. $\qquad\square$

### C.2 Proof of Theorem 2

This proof will again be completed in two parts: first for $p = 1$ (Lemma 3) and subsequently for $p > 1$ (Lemma 4).

**Lemma 3.** *For any $\beta \in (0, 1]$, the agent's best response is always a $\beta$-desirable effort profile if and only if there exists a desirable feature $f^\star$ ($f^\star \in \mathcal{D}$) such that:*

$$\max_{f \in \mathcal{U}} \frac{(\mathbb{C}h_0)_f}{c_f} < \frac{(\mathbb{C}h_0)_{f^\star}}{c_{f^\star}}.$$

*Proof.* We have already shown previously that there exists an optimal effort profile for the agent in which she needs to invest effort into a single feature $f^\star$ where

$$f^\star \in \arg\max_{f \in \mathcal{F}} \frac{(\mathbb{C}h_0)_f}{c_f},$$

and the optimal amount of effort that needs to be invested into that feature is given by $\frac{\alpha}{(\mathbb{C}h_0)_f}$. We define:

$$\mathcal{I}^* := \{f^\star \in \mathcal{F} : \ f^\star \in \arg\max_{f \in \mathcal{F}} \frac{(\mathbb{C}h_0)_f}{c_f}\}.$$

Note that $\mathcal{I}^* \neq \emptyset$ since $c > 0$, $(\mathbb{C}h_0) > 0$ and we have a finite number of features. Pick $f^\star \in \arg\max_{f \in \mathcal{D}} \frac{(\mathbb{C}h_0)_f}{c_f}$ ($f^\star$ exists and is in $\mathcal{D}$). Now, the agent's best response is a desirable effort profile if and only if:

$$
\begin{aligned}
\mathcal{I}^* \cap \mathcal{U} = \emptyset &\iff \forall f \in \mathcal{U}, \ f \notin \mathcal{I}^* \\
&\iff \forall f \in \mathcal{U}, \ \frac{(\mathbb{C}h_0)_f}{c_f} < \max_{f \in \mathcal{F}} \frac{(\mathbb{C}h_0)_f}{c_f} \\
&\iff \max_{f \in \mathcal{U}} \frac{(\mathbb{C}h_0)_f}{c_f} < \max_{f \in \mathcal{F}} \frac{(\mathbb{C}h_0)_f}{c_f} \\
&\iff \max_{f \in \mathcal{U}} \frac{(\mathbb{C}h_0)_f}{c_f} < \max_{f \in \mathcal{D}} \frac{(\mathbb{C}h_0)_f}{c_f} \\
&\iff \max_{f \in \mathcal{U}} \frac{(\mathbb{C}h_0)_f}{c_f} < \frac{(\mathbb{C}h_0)_{f^\star}}{c_{f^\star}}.
\end{aligned}
$$

This concludes the proof of the lemma. $\qquad\square$

**Lemma 4.** *For a $\ell_p$-norm cost function with $p > 1$, the agent's best response is always a $\beta$-desirable effort profile if and only if:*

$$\left[\sum_{f \in \mathcal{D}} \left(\frac{(\mathbb{C}h_0)_f}{c_f}\right)^{2/(p-1)}\right]^{1/2} \geq \frac{\beta}{\sqrt{1-\beta^2}} \left[\sum_{f \in \mathcal{U}} \left(\frac{(\mathbb{C}h_0)_f}{c_f}\right)^{2/(p-1)}\right]^{1/2}$$

*Proof.* We have already shown that when the agent's cost function is a $\ell_p$-norm with $p > 1$, her optimal effort profile $e^\star$ satisfies:

$$e_f^\star \propto \left(\frac{(\mathbb{C}h_0)_f}{c_f}\right)^{1/(p-1)} \qquad \forall f \in \mathcal{F}.$$

Now by the definition of $\beta$-desirability, $e^\star$ is $\beta$-desirable if and only if:

$$
\begin{aligned}
\|e_{\mathcal{D}}^\star\|_2 \geq \beta\|e^\star\|_2 &\iff \|e_{\mathcal{D}}^\star\|_2^2 \geq \frac{\beta^2}{1-\beta^2}\|e_{\mathcal{U}}^\star\|_2^2 \\
&\iff \sum_{f \in \mathcal{D}} e_f^{\star\,2} \geq \frac{\beta^2}{1-\beta^2} \sum_{f \in \mathcal{U}} e_f^{\star\,2} \\
&\iff \sum_{f \in \mathcal{D}} \left(\frac{(\mathbb{C}h_0)_f}{c_f}\right)^{2/(p-1)} \geq \frac{\beta^2}{1-\beta^2} \sum_{f \in \mathcal{U}} \left(\frac{(\mathbb{C}h_0)_f}{c_f}\right)^{2/(p-1)} \\
&\iff \left[\sum_{f \in \mathcal{D}} \left(\frac{(\mathbb{C}h_0)_f}{c_f}\right)^{2/(p-1)}\right]^{1/2} \geq \frac{\beta}{\sqrt{1-\beta^2}} \left[\sum_{f \in \mathcal{U}} \left(\frac{(\mathbb{C}h_0)_f}{c_f}\right)^{2/(p-1)}\right]^{1/2}.
\end{aligned}
$$

This concludes the proof of the lemma. $\qquad\square$

### C.3 Proof of Theorem 3

We will complete the proof in two parts. First, we will show that in the general case, the space of desirable classifiers $\mathcal{H}$ can be non-convex (Prop 3), followed by the proof of convexity in the special case of $|\mathcal{D}| = 1$ (Prop 4).

**Proposition 3.** *For any $p \geq 1$, there always exists an instance of the problem $(\mathbb{C}, h_0)$ and $\beta > 0$ such that the space of $\beta$-desirable classifiers $\mathcal{H}$ is a non-convex set.*

*Proof.* The set of $\beta$-desirable classifiers $\mathcal{H}$ is given as follows:

$$\mathcal{H} := \left\{ h_0 \in \mathbb{R}^{|\mathcal{F}|} : e^\star(h_0, \mathbb{C}) \text{ is } \beta\text{-desirable}, \mathbb{C}h_0 \geq 0 \right\}.$$

We define the set $\mathcal{Z} := \{(\mathbb{C}h_0) : h_0 \in \mathcal{H}\}$. Suppose that $\mathbb{C}$ is full row-rank. This implies that $\mathcal{H}$ is convex if and only if $\mathcal{Z}$ is convex[12]. Therefore, in order to complete the proof, it suffices to show that the transformed set $\mathcal{Z}$ is non-convex in the worst case. We now provide instances of problems where $\mathcal{Z}$ is non-convex and the agents incur $\ell_p$-norm cost functions with $p = 1$ and $p > 1$. Recall from Theorem 2 that the set $\mathcal{Z}$ is given as follows:

$$\mathcal{Z} = \left\{ z \in \mathbb{R}^{|\mathcal{F}|}_{\geq 0} : \max_{f \in \mathcal{U}} \frac{z_f}{c_f} < \max_{f \in \mathcal{D}} \frac{z_f}{c_f} \right\} \quad (p = 1)$$

$$\mathcal{Z} = \left\{ z \in \mathbb{R}^{|\mathcal{F}|}_{\geq 0} : \left[ \sum_{f \in \mathcal{D}} \left( \frac{z_f}{c_f} \right)^{2/(p-1)} \right]^{1/2} \geq \frac{\beta}{\sqrt{1-\beta^2}} \left[ \sum_{f \in \mathcal{U}} \left( \frac{z_f}{c_f} \right)^{2/(p-1)} \right]^{1/2} \right\} \quad (p > 1)$$

**Weighted $\ell_1$-norm cost function:** Consider a setting where there are 4 features with $\mathcal{D} = \{1, 2\}$ and $\mathcal{U} = \{3, 4\}$. Suppose that the cost vector equals $c = \mathbf{1}$. In this case,

$$\mathcal{Z} = \left\{ z \in \mathbb{R}^4_{\geq 0} : \max(z_3, z_4) < \max(z_1, z_2) \right\}.$$

Now, choose $z' := (4, 7, 3, 6)$ and $z'' := (7, 4, 3, 6)$. Both are clearly points in $\mathcal{Z}$. However, for $\alpha = 0.5$, $\alpha z' + (1 - \alpha)z'' := (5.5, 5.5, 3, 6) \notin \mathcal{Z}$. Therefore, $\mathcal{Z}$ is not a convex set.

**Weighted $\ell_p$-norm cost function:** Consider a setting where there are 3 features with $\mathcal{D} = \{1, 2\}$ and $\mathcal{U} = \{3\}$. Let $p = 2$, $c = \mathbf{1}$ and $\beta = \frac{1}{\sqrt{2}}$. Then $\mathcal{Z}$ is given by:

$$\mathcal{Z} = \left\{ z \in \mathbb{R}^3_{\geq 0} : \sqrt{z_1^2 + z_2^2} \geq z_3 \right\}$$

$(0, 1, 1)$ and $(1, 0, 1)$ are points in $\mathcal{Z}$, but the point halfway between them, given by $(0.5, 0.5, 1)$ is clearly not in $\mathcal{Z}$. Therefore, $\mathcal{Z}$ is not a convex set. This concludes the proof. $\square$

**Proposition 4.** *Suppose that there is only a single desirable feature, i.e., that $|\mathcal{D}| = 1$. Then for any $\beta > 0$, the space of $\beta$-desirable classifiers $\mathcal{H}$ is convex for any $\ell_p$-norm cost function with $p \in [1, 3]$.*

*Proof.* We will verify convexity separately for the cases with $\ell_1$-norm and $\ell_p$-norm ($1 < p \leq 3$) cost functions.

*The $\ell_1$-norm case:* When the cost function is a weighted $\ell_1$-norm, the set of desirable classifiers is given by

$$\mathcal{H} := \left\{ h_0 \in \mathbb{R}^{|\mathcal{F}|} : \mathbb{C}h_0 \geq 0, \max_{f \in \mathcal{U}} \frac{(\mathbb{C}h_0)_f}{c_f} < \max_{f \in \mathcal{D}} \frac{(\mathbb{C}h_0)_f}{c_f} \right\}.$$

Now, suppose $|\mathcal{D}| = 1$ and there is some feature $f_d \in \mathcal{D}$. Then, we can rewrite $\mathcal{H}$ as follows:

$$\mathcal{H} := \left\{ h_0 \in \mathbb{R}^{|\mathcal{F}|} : \mathbb{C}h_0 \geq 0, \max_{f \in \mathcal{F} \setminus \{f_d\}} \frac{(\mathbb{C}h_0)_f}{c_f} - \frac{(\mathbb{C}h_0)_{f_d}}{c_{f_d}} < 0 \right\}.$$

In order to show that $\mathcal{H}$ is a convex set, it suffices to show that the function $g(h_0) = \max_{f \in \mathcal{F} \setminus \{f_d\}} \frac{(\mathbb{C}h_0)_f}{c_f} - \frac{(\mathbb{C}h_0)_{f_d}}{c_{f_d}}$ is a convex function. Function $g(\cdot)$ corresponds to the sum of a maximum of linear functions (which is convex) and a linear function; hence, function $g(\cdot)$ is convex.

*The $\ell_p$-norm case with $p > 1$:* For $\ell_p$-norm cost functions with $p > 1$, set $\mathcal{H}$ is given by:

$$\mathcal{H} \triangleq \left\{ h_0 \in \mathbb{R}^{|\mathcal{F}|} : \mathbb{C}h_0 \geq 0, \left[ \sum_{f \in \mathcal{D}} \left( \frac{(\mathbb{C}h_0)_f}{c_f} \right)^{2/(p-1)} \right]^{1/2} \geq \frac{\beta}{\sqrt{1-\beta^2}} \left[ \sum_{f \in \mathcal{U}} \left( \frac{(\mathbb{C}h_0)_f}{c_f} \right)^{2/(p-1)} \right]^{1/2} \right\}$$

---

[12]This is a standard result in linear algebra; the proof provided in Appendix E for completeness

Using the fact that $|\mathcal{D}| = 1$, we can rewrite $\mathcal{H}$ as follows:

$$\mathcal{H} := \left\{ h_0 \in \mathbb{R}^{|\mathcal{F}|} : \mathbb{C}h_0 \geq 0, (\mathbb{C}h_0)_{f_d} \geq K \left[ \sum_{f \in \mathcal{U}} \left( \frac{(\mathbb{C}h_0)_f}{c_f} \right)^{2/(p-1)} \right]^{(p-1)/2} \right\}$$

where $K = c_{f_d} \left( \frac{\beta}{\sqrt{1-\beta^2}} \right)^{(p-1)} > 0$. Now in order to complete the proof, we need to show that the function $r(h_0)$ is convex, where $r(h_0)$ is given by:

$$r(h_0) = K \left[ \sum_{f \in \mathcal{U}} \left( \frac{(\mathbb{C}h_0)_f}{c_f} \right)^{2/(p-1)} \right]^{(p-1)/2} - (\mathbb{C}h_0)_{f_d}.$$

When $1 < p \leq 3$, we can rewrite $r(h_0)$ as follows:

$$r(h_0) = K\|Bh_0\|_{2/(p-1)} - (\mathbb{C}h_0)_{f_d},$$

where $B \in \mathbb{R}^{(|\mathcal{F}|-1)\times(|\mathcal{F}|-1)}$. Note that $K\|Bh_0\|_{2/(p-1)}$ is a convex function in $h_0$ since this is a $q$-norm for $q = \frac{2}{p-1} \geq 1$. This makes $r(h_0)$ a convex function in $h_0$ (sum of a convex function and a linear function is convex) and concludes the proof. $\qquad\square$

### C.4 Heuristic for Inducing Desirable Effort when $|\mathcal{D}| > 1$

When $|\mathcal{D}| > 1$, we know that in general, the set $\mathcal{H}$ of $\beta$-desirable classifiers is not convex, and difficult to optimize over. However, we propose a convexification heuristic (parameterized by $\gamma$) where the principal just tries to design a classifier such that "the total contribution of undesirable features is no more than $\gamma$":

**Proposition 5.** *Let $\mathcal{H}_{w(\mathcal{U})\leq\gamma} = \{h_0 : \|(\mathbb{C}h_0)_{\mathcal{U}}\|_{2/(p-1)} \leq \gamma\}$. Then for any $\gamma > 0$, $\mathcal{H}_{w(\mathcal{U})\leq\gamma}$ is convex for any $\ell_p$-norm cost function with $p \in [1, 3]$.*

The proof is nearly identical to that of Proposition 4 and is omitted to avoid repetition. This result helps the principal guarantee that they can bound the effort exerted on undesirable features, even if they are not able to guarantee a certain target level of $\beta$-desirable effort.

### C.5 Hardness Results

In this segment, we try to formally characterize hardness of: a) finding any $\beta$-desirable classifier, and b) finding the **best** such classifier, i.e., one which is simultaneously $\beta$-desirable and also maximizes classification accuracy. We operate in the limited setting of $\ell_2$-norm cost functions. We show that while subproblem a) can be solved in polynomial time, subproblem b) is likely to be NP-hard.

**a) It is possible to find a $\beta$-desirable classifier for $\ell_2$-norm cost functions in polynomial time.**

Consider the matrices $C_\mathcal{D}$ and $C_\mathcal{U}$, where $C_\mathcal{D}$ zeroes out all rows not corresponding to features in $\mathcal{D}$, and $C_\mathcal{U}$ zeroes out all rows not corresponding to features in $\mathcal{U}$, from the contribution matrix $C$. Theorem 2 shows that the problem of finding desirable classifiers reduces to (when $c = 1, p = 2$) finding an $h$ such that

$$\left( \frac{\|C_\mathcal{D}h\|}{\|C_\mathcal{U}h\|} \right)^2 \geq \frac{\beta^2}{1-\beta^2},$$

or equivalently

$$\frac{h^\top(C_\mathcal{D}^\top C_\mathcal{D})h}{h^\top C_\mathcal{U}^\top C_\mathcal{U}h} \geq \frac{\beta^2}{1-\beta^2}.$$

To find a feasible solution, it suffices to find a value of $h$ that maximizes the LHS. Because both $C_\mathcal{D}^\top C_\mathcal{D}$ and $C_\mathcal{U}^\top C_\mathcal{U}$ are positive-semidefinite, maximizing the LHS is a known problem with polynomial-time algorithms; we are maximizing a generalized Rayleigh quotient, and the optimal $h$ is the eigenvector for the maximum eigenvalue of a well-specified function of matrices $A$ and $B$.

**b) Finding the *best* such classifier is at least as hard as certifying feasibility of the set of constraints $A^\top h \geq b$ and $h^\top Q h \geq 0$ where Q is diagonal and has both positive and negative eigenvalues.**

We will focus on the version of the problem where, given observations $\{(x_i, y_i)\}_{i=1}^N$ where $x_i$ is a $d$-dimensional vector and $y_i$ is a real-valued label, we aim to find the classifier $h$ that is $\beta$-desirable and minimizes the worst-case error across all data samples, i.e., it minimizes

$$f(h) = \|X^\top h - Y\|_\infty,$$

where $X$ is the data matrix and $Y$ the label vector. We make the assumption that $C = I_d$ is diagonal, i.e., we want to show that our hardness result holds even in the simplest causal graph setting where features do not affect each other. We first prove an intermediate result. Let $\mathcal{H}_\beta$ be the space of all $\beta$-desirable classifiers.

**Lemma 5.** *Solving the optimization problem:* $\min_{h \in \mathcal{H}_\beta} f(h) = \|X^\top h - Y\|_\infty$ *is at least as hard as checking the feasibility, given any $\delta > 0$, of a problem of the following problem:*

$$Y - \delta \cdot \mathbf{1} \leq X^\top h,$$
$$h^\top Q' h \geq 0,$$

*where $Q'$ is a diagonal matrix with two distinct eigenvalues $\lambda > 0$ and $\lambda' < 0$.*

*Proof.* For the given optimization problem, the problem is at least as hard as deciding for any given $\delta > 0$, whether the optimal solution is $\leq \delta$ or $> \delta$. We can rewrite this as the following feasibility problem:

*Given $\delta$, does there exist a classifier $h$ such that the following conditions hold:*

$$\|X^\top h - Y\|_\infty \leq \delta, \quad \text{and}$$

$$\|h_\mathcal{D}\| \geq \frac{\beta}{\sqrt{1 - \beta^2}} \|h_\mathcal{U}\|.$$

Note that

$$\|X^\top h - Y\|_\infty \leq \delta \iff \max_{i \in [N]} \left| x_i^\top h - y_i \right| \leq \delta$$

$$\iff \left| x_i^\top h - y_i \right| \leq \delta \quad \forall i \in [N]$$

$$\iff -\delta \leq x_i^\top h - y_i \leq \delta \quad \forall i \in [N]$$

$$\iff -\delta \cdot \mathbf{1} \leq X^\top h - Y \leq \delta \cdot \mathbf{1}.$$

Now, define $I_\mathcal{D} = diag([\mathbf{1}_{|\mathcal{D}|}, \mathbf{0}_{|\mathcal{U}|}])$ and $I_\mathcal{U} = diag([\mathbf{0}_{|\mathcal{D}|}, \mathbf{1}_{|\mathcal{U}|}])$. Then, for the other constraint we have:

$$\|h_\mathcal{D}\| \geq \frac{\beta}{\sqrt{1 - \beta^2}} \|h_\mathcal{U}\| \iff \|I_\mathcal{D} h\| \geq \frac{\beta}{\sqrt{1 - \beta^2}} \|I_\mathcal{U} h\|$$

$$\iff h^\top I_\mathcal{D} h \geq \frac{\beta^2}{1 - \beta^2} h^\top I_\mathcal{U} h \quad (\text{note: } I_\mathcal{D}^\top I_\mathcal{D} = I_\mathcal{D}, I_\mathcal{U}^\top I_\mathcal{U} = I_\mathcal{U})$$

$$\iff h^\top Q' h \geq 0,$$

where $Q'$ be the diagonal matrix with entries 1 for features in $\mathcal{D}$, and $\frac{-\beta^2}{1-\beta^2}$ for features in $\mathcal{U}$. Putting it all together, we have the following equivalent feasibility problem:

$$-\delta \cdot \mathbf{1} \leq X^\top h - Y \leq \delta \cdot \mathbf{1}, \quad \text{and}$$
$$h^\top Q' h \geq 0,$$

This is at least as hard as solving the feasibility problem on the superset

$$-\delta \cdot \mathbf{1} \leq X^\top h - Y, \quad \text{and}$$
$$h^\top Q' h \geq 0,$$

(since feasibility on the subset implies feasibility on the superset). This concludes the proof. $\qquad \square$

According to a classical optimization result, the class of feasibility problems of the form: $A^\top h \geq b$ and $h^\top Q h \geq \delta$ where $Q$ is diagonal but neither positive or negative semi-definite, is NP-hard (see hardness results of this form surveyed in [34]). We do note a small gap here however: while changing $X$, $Y$ and $\lambda$ allows us to generate the entire set of constraints of the form $Ax \geq b$, $Q'$ cannot be used to generate the set of *all* diagonal matrices $Q$ because $Q'$ only has two distinct eigenvalues. We technically do not map to the entire NP-hard class of problems of the form $x^\top Q x \geq \delta \ \& \ A^\top x \geq b$, but we this provides evidence that the problem is likely NP-hard.

Finally, we note that the general consensus in the optimization community is that high-dimensional non-convex optimization problems are considered intractable, and there are no practical/general-purpose methods to solve them.

# D   Results & Proofs for the Incomplete Information Setting

## D.1   Proof of Theorem 4

We will complete the proof as follows: first, we will argue that for scenarios (a) and (b), $\mathbb{C}h$ is Gaussian. Then, we will show in Lemma 6 that for $\mathbb{C}h$ Gaussian, Problem (4) is indeed a convex program and can be reduced to Problem (5). Finally, we will show in Lemma 7 that for all other scenarios, the problem is non-convex.

In scenario (a), when the causal graph $\mathcal{G}$ is fully known and there is uncertainty only over the classifier, it is clear that each entry of $\mathbb{C}h$ is a linear combination of Gaussian random variables and therefore, $\mathbb{C}h$ is Gaussian. In scenario (b), since the causal graph is bi-partite with all arcs oriented in the same direction, any feature $i$ may affect any other feature $j$ either directly or not at all. Therefore, entry $\mathbb{C}_{ij}$ is either $0$ or $w(a_{ij})$ which is a Gaussian random variable. Since the classifier is fully known, $\mathbb{C}h$ is again multi-variate Gaussian.

**Lemma 6.** *Under any partially incomplete information model where $\mathbb{C}h$ is multi-variate Gaussian and for cost functions given by Eq. (1), the agent's optimization problem to find the optimal effort profile $e^\star$, given by (4), is a convex program for any $\delta \leq \frac{1}{2}$.*

*Proof.* Since the cost functions defined in Eq. (1) are convex, in order to complete the proof, it suffices to show that the feasible space of the optimization problem in (4), is convex when $\mathbb{C}h$ is multi-variate Gaussian. Suppose, $\mathbb{C}h \sim \mathcal{N}(\mu_{\mathbb{C}h}, \Sigma_{\mathbb{C}h})$ for some $\mu_{\mathbb{C}h} \in \mathbb{R}^{|\mathcal{F}|}$ and $\Sigma_{\mathbb{C}h} \in \mathbb{R}^{|\mathcal{F}| \times |\mathcal{F}|}$. This implies,

$$(\mathbb{C}h)^\top e \sim \mathcal{N}\left(\mu_{\mathbb{C}h}^\top e, e^\top \Sigma_{\mathbb{C}h} e\right).$$

This allows us to rewrite the LHS of the probability constraint as follows:

$$\mathbb{P}\left[(\mathbb{C}h)^\top e \geq \alpha\right] = \mathbb{P}\left[\frac{(\mathbb{C}h)^\top e - \mu_{\mathbb{C}h}^\top e}{\sqrt{e^\top \Sigma_{\mathbb{C}h} e}} \geq \frac{\alpha - \mu_{\mathbb{C}h}^\top e}{\sqrt{e^\top \Sigma_{\mathbb{C}h} e}}\right]$$

$$= \mathbb{P}\left[Z \geq \frac{\alpha - \mu_{\mathbb{C}h}^\top e}{\sqrt{e^\top \Sigma_{\mathbb{C}h} e}}\right] \quad \text{(where } Z \sim \mathcal{N}(0, 1))$$

$$= \Phi^c\left(\frac{\alpha - \mu_{\mathbb{C}h}^\top e}{\sqrt{e^\top \Sigma_{\mathbb{C}h} e}}\right).$$

Therefore,

$$\mathbb{P}\left[(\mathbb{C}h)^\top e \geq \alpha\right] \geq 1 - \delta \iff \Phi^c\left(\frac{\alpha - \mu_{\mathbb{C}h}^\top e}{\sqrt{e^\top \Sigma_{\mathbb{C}h} e}}\right) \geq 1 - \delta$$

$$\iff \Phi\left(\frac{\alpha - \mu_{\mathbb{C}h}^\top e}{\sqrt{e^\top \Sigma_{\mathbb{C}h} e}}\right) \leq \delta$$

$$\iff \frac{\alpha - \mu_{\mathbb{C}h}^\top e}{\sqrt{e^\top \Sigma_{\mathbb{C}h} e}} \leq p_\delta \quad \text{(where } p_\delta = \Phi^{-1}(\delta))$$

$$\iff \alpha - \mu_{\mathbb{C}h}^\top e - p_\delta \cdot \sqrt{e^\top \Sigma_{\mathbb{C}h} e} \leq 0.$$

When $\delta = \frac{1}{2}$, $p_\delta = 0$ and the above constraint reduces to a polyhedral constraint, making the problem trivially convex. On the other hand, note that when $\delta < \frac{1}{2}$, $p_\delta < 0$. Now, since $\Sigma_{\mathbb{C}h}$ is a covariance matrix, it is always symmetric and positive semidefinite and therefore, $\Sigma_{\mathbb{C}h}^{1/2}$ exists (it is also symmetric and positive semidefinite!). In that case, we can express $\sqrt{e^\top \Sigma_{\mathbb{C}h} e}$ as follows:

$$\sqrt{e^\top \Sigma_{\mathbb{C}h} e} = \sqrt{e^\top \Sigma_{\mathbb{C}h}^{1/2} \Sigma_{\mathbb{C}h}^{1/2} e} = \sqrt{(\Sigma_{\mathbb{C}h}^{1/2} e)^\top (\Sigma_{\mathbb{C}h}^{1/2} e)} = \sqrt{||\Sigma_{\mathbb{C}h}^{1/2} e||_2^2} = ||\Sigma_{\mathbb{C}h}^{1/2} e||_2.$$

Now, $||\Sigma^{1/2} e||_2$ is a convex function in $e$ (because all norms are convex functions). Similarly, $-p_\delta \cdot ||\Sigma_{\mathbb{C}h}^{1/2} e||_2$ is also a convex function because $-p_\delta > 0$. The term $\alpha - \mu_{\mathbb{C}h}^\top e$ is affine in $e$ and therefore, convex by default. Putting everything together, we conclude that $\alpha - \mu_{\mathbb{C}h}^\top e - p_\delta \cdot \sqrt{e^\top \Sigma_{\mathbb{C}h} e}$ is a convex function in $e$ which makes the constraint:

$$\alpha - \mu_{\mathbb{C}h}^\top e - p_\delta \cdot \sqrt{e^\top \Sigma_{\mathbb{C}h} e} \leq 0$$

a convex constraint. This concludes the proof of the lemma. $\qquad\square$

**Lemma 7.** *Under incomplete information model* (3) *and general cases of model* (2) *where the causal graph is not bipartite, the agent's optimization problem, given by* (4)*, is a non-convex program.*

*Proof.* In order to complete this proof, it suffices to provide counter-examples where the program in (4) is non-convex.

**Model 3.** Consider the simplest possible setting where there can be uncertainty in both the classifier and the causal graph. Suppose, there is only one feature, i.e., $|\mathcal{F}| = 1$. Let $\omega \sim \mathcal{N}(0,1)$ be the random variable that captures the uncertainty in the contribution of the feature (encodes uncertainty in the causal graph) and $h \sim \mathcal{N}(0,1)$ be the random variable that captures the uncertainty in the classifier weight on the feature. Note that $\omega \perp h$. We will show that the feasible space given by:

$$\mathbb{P}\left[(\omega h)e \geq \alpha\right] \geq 1 - \delta$$

is non-convex, which is equivalent to showing that the function $f(e)$ given by:

$$f(e) = \mathbb{P}\left[(\omega h)e \geq \alpha\right]$$

is not concave. Below in Figure 6, we plot $f(e)$ as a function of one-dimensional effort $e$. Since there is no closed-form expression for the distribution of the product of two independent standard normal random variables, we obtain empirical estimates for the probability at each $e$ using Monte-Carlo simulations. Clearly, $f(e)$ is not concave.

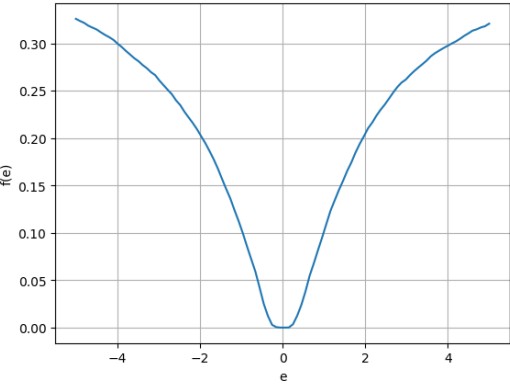

Figure 6: Plot of $f(e)$ with $\alpha = 1$

**General Cases of Model** 2 **with Non-bipartite Graphs.** Consider any non-bipartite causal graph. There must exist a pair of features $i$ and $j$ such that $i$ affects $j$ indirectly. In that case, the entry $\mathbb{C}_{ij}$ of

the contribution matrix must contain at least one term which is a product of multiple independent Gaussian random variables. Therefore, even when the classifier is fully known, by a similar argument as Model 3, the optimization problem is again non-convex.

This concludes the proof of the lemma. □

## D.2 Characterization of Feasibility of Problem 5

**Proposition 6.** *Suppose that $\Sigma_{\mathbb{C}h}$ is positive definite. Then the optimization problem* (5) *is feasible if and only if $\delta > \Phi^{-1}\left(-\|\Sigma_{\mathbb{C}h}^{-1/2}\mu_{\mathbb{C}h}\|_2\right)$, where $\Phi^{-1}(\cdot)$ indicates the inverse of the standard normal CDF.*

*Proof.* Recall that the feasible space of the agent's optimization problem in the partially incomplete information case is given by:

$$\alpha - \mu_{\mathbb{C}h}^\top e - p_\delta \cdot \|\Sigma_{\mathbb{C}h}^{1/2} e\|_2 \le 0.$$

This feasible space is empty at a given $\delta$ if we have:

$$\alpha - \mu_{\mathbb{C}h}^\top e - p_\delta \cdot \|\Sigma_{\mathbb{C}h}^{1/2} e\|_2 > 0 \quad \forall\, e \iff \min_e \quad g(e) = \alpha - \mu_{\mathbb{C}h}^\top e - p_\delta \cdot \|\Sigma_{\mathbb{C}h}^{1/2} e\|_2 > 0.$$

Now consider the convex unconstrained optimization problem: $\min_e g(e)$. Then,

$$\nabla g(e) = -\mu_{\mathbb{C}h} - p_\delta \cdot \frac{\Sigma_{\mathbb{C}h} e}{\|\Sigma_{\mathbb{C}h}^{1/2} e\|_2}.$$

Now there are 2 cases: $\nabla g(e) = 0$ *has a solution $\hat{e}$:* Clearly, $\hat{e} \ne 0$ because the gradient is not defined at $e = 0$. In that case, $\hat{e}$ satisfies:

$$-p_\delta \cdot \frac{\Sigma_{\mathbb{C}h} \hat{e}}{\|\Sigma_{\mathbb{C}h}^{1/2} \hat{e}\|_2} = \mu_{\mathbb{C}h}.$$

Now, $\hat{e}$ must be a global minimizer of $g$ because $g(\cdot)$ is a convex function. We will show that $g(\hat{e}) = \alpha$:

$$\begin{aligned}
g(\hat{e}) &= \alpha - \mu_{\mathbb{C}h}^\top \hat{e} - p_\delta \cdot \|\Sigma_{\mathbb{C}h}^{1/2} \hat{e}\|_2 \\
&= \alpha + p_\delta \cdot \frac{\hat{e}^\top \Sigma_{\mathbb{C}h} \hat{e}}{\|\Sigma_{\mathbb{C}h}^{1/2} \hat{e}\|_2} - p_\delta \cdot \|\Sigma_{\mathbb{C}h}^{1/2} \hat{e}\|_2 \quad \text{(using the condition from } \nabla g(\hat{e}) = 0\text{)} \\
&= \alpha + p_\delta \cdot \|\Sigma_{\mathbb{C}h}^{1/2} \hat{e}\|_2 - p_\delta \cdot \|\Sigma_{\mathbb{C}h}^{1/2} \hat{e}\|_2 \\
&= \alpha.
\end{aligned}$$

Since $\alpha > 0$, the problem is always infeasible for this particular value of $p_\delta$. Since $\Sigma_{\mathbb{C}h}$ is positive definite, $\Sigma_{\mathbb{C}h}^{-1/2}$ exists, therefore we have:

$$p_\delta = -\|\Sigma_{\mathbb{C}h}^{-1/2} \mu_{\mathbb{C}h}\|_2 \iff \delta = \Phi^{-1}\left(-\|\Sigma_{\mathbb{C}h}^{-1/2} \mu_{\mathbb{C}h}\|_2\right).$$

$\nabla g(e) = 0$ *has no solution:* This means that either the unique optimal solution is at the point where the gradient does not exist, i.e, $e = 0$, or the solution is unbounded. The first subcase clearly leads to infeasibility as $g(0) = \alpha > 0$ while the second sub-case leads to a non-empty feasible region for Problem (5). We will now try to derive conditions on $\delta$ which lead to each subcase.

Suppose that the unique optimal solution is $e = 0$. This means that for any direction $d$, $g(0+d) > g(0)$ or equivalently,

$$\alpha - \mu_{\mathbb{C}h}^\top d - p_\delta \cdot \|\Sigma_{\mathbb{C}h}^{1/2} d\|_2 > \alpha \quad \forall\, d \iff -p_\delta \cdot \|\Sigma_{\mathbb{C}h}^{1/2} d\|_2 > \mu_{\mathbb{C}h}^\top d \quad \forall\, d.$$

Note that $\mu_{\mathbb{C}h}^\top d = \mu_{\mathbb{C}h}^\top \Sigma_{\mathbb{C}h}^{-1/2} \Sigma_{\mathbb{C}h}^{1/2} d = (\Sigma_{\mathbb{C}h}^{-1/2} \mu_{\mathbb{C}h})^\top (\Sigma_{\mathbb{C}h}^{1/2} d) \le \|\Sigma_{\mathbb{C}h}^{-1/2} \mu_{\mathbb{C}h}\|_2 \cdot \|\Sigma_{\mathbb{C}h}^{1/2} d\|_2$ where the last inequality follows from the Cauchy-Schwartz inequality. In fact, for $d^* = \Sigma_{\mathbb{C}h}^{-1} \mu_{\mathbb{C}h}$ (which exists

since $\Sigma_{\mathbb{C}h}$ is positive definite and hence, invertible), we have equality. But since $-p_\delta \cdot \|\Sigma_{\mathbb{C}h}^{1/2} d\|_2 > \mu_{\mathbb{C}h}^\top d$ for all directions $d$, it must hold for $d^*$ as well, which implies:

$$\|\Sigma_{\mathbb{C}h}^{-1/2} \mu_{\mathbb{C}h}\|_2 \cdot \|\Sigma_{\mathbb{C}h}^{1/2} d^*\|_2 < -p_\delta \cdot \|\Sigma_{\mathbb{C}h}^{1/2} d^*\|_2,$$

which means that $-p_\delta > \|\Sigma_{\mathbb{C}h}^{-1/2} \mu_{\mathbb{C}h}\|_2$ or equivalently, $\delta < \Phi^{-1}\left(-\|\Sigma_{\mathbb{C}h}^{-1/2} \mu_{\mathbb{C}h}\|_2\right)$.

Similarly, if the solution is unbounded, there must exist a direction $d'$ at 0 such that:

$$\alpha - \mu_{\mathbb{C}h}^\top d' - p_\delta \cdot \|\Sigma_{\mathbb{C}h}^{1/2} d'\|_2 < \alpha,$$

or equivalently, $-p_\delta \cdot \|\Sigma_{\mathbb{C}h}^{1/2} d'\|_2 < \mu_{\mathbb{C}h}^\top d'$. Using a similar argument as above, we can show that this can happen only when:

$$\delta > \Phi^{-1}\left(-\|\Sigma_{\mathbb{C}h}^{-1/2} \mu_{\mathbb{C}h}\|_2\right).$$

This concludes the proof of the proposition. $\qquad\square$

### D.3 Structure of Optimal Effort Profiles under Weighted $\ell_1$-norm Costs

**Proposition 7.** *Under any partially incomplete information setting which leads to the convex optimization problem* (5) *for the agent, the optimal effort profile $e^\star$ with weighted $\ell_1$-norm costs requires investment of effort into more than one feature in the worst case.*

*Proof.* In order to complete the proof, it suffices to construct an instance of the problem where the optimal effort profile is not a corner point. Consider a setting where $|\mathcal{F}| = 2$, $\alpha > 0$ and $\delta < \frac{1}{2}$. Suppose, the features are identical in all respects, i.e., $(\mu_{\mathbb{C}h})_1 = (\mu_{\mathbb{C}h})_2 = \bar{\mu} > 0$, $\Sigma_{\mathbb{C}h} = \begin{bmatrix} \sigma^2 & 0 \\ 0 & \sigma^2 \end{bmatrix}$ and $c_1 = c_2 = c$. Additionally, suppose that $\bar{\mu} > -p_\delta \sigma$. We first make the following observations:

- If $e^\star$ is not a corner point, it must be symmetric, i.e., $e_1^\star = e_2^\star$.

- Since $\bar{\mu} > 0$ and $\delta < \frac{1}{2}$, it must be that $e^\star \geq 0$ (otherwise, we have infeasibility).

Now, there are only two possible corner point solutions: either of the form $(e, 0)$ or $(0, e)$. In order for either of them to be optimal, the constraint must be active at that point. Solving, we obtain:

$$e = \frac{\alpha}{\bar{\mu} + p_\delta \sigma}; \quad \text{and} \quad \text{Cost} = \frac{c\alpha}{\bar{\mu} + p_\delta \sigma}.$$

However, we will now construct a non-corner point solution $(e', e')$ where the constraint is active and which produces a strictly better objective value. Solving, we obtain:

$$e' = \frac{\alpha}{2\left(\bar{\mu} + \frac{p_\delta}{\sqrt{2}} \cdot \sigma\right)}; \quad \text{and} \quad \text{Cost} = \frac{c\alpha}{\left(\bar{\mu} + \frac{p_\delta}{\sqrt{2}} \cdot \sigma\right)},$$

which is strictly smaller than the earlier cost (since $p_\delta < 0$). This concludes the proof. $\qquad\square$

### D.4 Proof of Theorem 5

We will use the KKT conditions to obtain the agent's optimal effort profile $e^\star$. The Lagrangian $\mathcal{L}(\cdot, \cdot)$ for the above problem is given by:

$$\mathcal{L}(e, \lambda) = \|e\|_2 + \lambda\left(\alpha - \mu_{\mathbb{C}h}^\top e - p_\delta \cdot \|\Sigma_{\mathbb{C}h}^{1/2} e\|_2\right),$$

where $\lambda$ is the Lagrange multiplier. We can now write the KKT conditions as follows:

$$\frac{e}{\|e\|_2} + \lambda \cdot \left(-p_\delta \cdot \frac{\Sigma_{\mathbb{C}h} e}{\|\Sigma_{\mathbb{C}h}^{1/2} e\|_2} - \mu_{\mathbb{C}h}\right) = 0,$$
$$p_\delta \cdot \|\Sigma_{\mathbb{C}h}^{1/2} e\|_2 + \mu_{\mathbb{C}h}^T e = \alpha,$$
$$\lambda > 0.$$

Since we have a convex program, it is sufficient to find a pair $(e^\star, \lambda^\star)$ satisfying the KKT conditions and we can immediately conclude that $e^\star$ is an optimal solution to our original problem. Using the first equality above, we infer that the optimal effort $e^\star$ must be of the following form:

$$e^\star = \lambda^* \left(k_1 I + k_2 \Sigma_{\mathbb{C}h}\right)^{-1} \mu,$$

where $k_1 = \frac{1}{||e^\star||_2} > 0$ and $k_2 = \frac{-\lambda^* p_\delta}{||\Sigma_{\mathbb{C}h}^{1/2} e^\star||_2} > 0$ (so, the inverse exists). In order to obtain the exact expression for $e^\star$, we need to use the other equality condition and solve simultaneously for $k_1$, $k_2$ and $\lambda^*$.

The last part of the theorem follows directly by noting that $\Sigma_{\mathbb{C}h}$ is a diagonal matrix. In that case, $(k_1 I + k_2 \Sigma_{\mathbb{C}h})^{-1}$ is also a diagonal matrix where the diagonal entry corresponding to feature $f$ is given by $1/(k_1 + k_2 (\Sigma_{\mathbb{C}h})_f)$. The final expression follows from simple algebra. This concludes the proof of the theorem.

### D.5 Scenario where $\Sigma_{\mathbb{C}h}$ is a diagonal matrix

**Proposition 8.** *Suppose $\mathcal{G}$ is a bipartite graph with all edges oriented in the same direction; further, suppose the agent only has uncertainty over the weights of the graph (model 2), i.e. $\Sigma_h = \mathbf{0}$. Then, $\Sigma_{\mathbb{C}h}$ is a diagonal matrix.*

*Proof.* Since $\mathcal{G}$ is a bipartite graph, the set of nodes (in this case, same as set of features) $|\mathcal{F}|$ can be partitioned into two sets $\mathcal{F}_{out}$ and $\mathcal{F}_{in}$ such that $\mathcal{F}_{in} \cup \mathcal{F}_{out} = \mathcal{F}$, $\mathcal{F}_{in} \cap \mathcal{F}_{out} = \emptyset$ and all arcs in $\mathcal{A}$ are directed from $\mathcal{F}_{out}$ towards $\mathcal{F}_{in}$.

Recall that $\Sigma_{\mathbb{C}h}$ is the covariance matrix of $\mathbb{C}h$ where $\mathbb{C} \sim \Pi_{\mathbb{C}}$ and $h \sim \Pi_h$. However, when there is uncertainty only over the edge weights of $\mathcal{G}$, it is clear that $\Sigma_{\mathbb{C}h} = Cov(\mathbb{C}h_0)$. Therefore, in order to show that $\Sigma_{\mathbb{C}h}$ is a diagonal matrix, it suffices to show that:

$$\forall\, f_1, f_2 \in \mathcal{F}, f_1 \neq f_2, \quad (\mathbb{C}h_0)_{f_1} \perp (\mathbb{C}h_0)_{f_2},$$

i.e., $(\mathbb{C}h_0)_{f_1}$ and $(\mathbb{C}h_0)_{f_2}$ are independent random variables. Firstly, observe that for any feature $f \in \mathcal{F}_{in}$, we must have:

$$(\mathbb{C}h_0)_f = 0.$$

This is because feature $f$ has no outgoing edges (since $f \in \mathcal{F}_{in}$) and therefore, $\mathbb{C}_{f,.} = \mathbf{0}^\top$ which implies $(\mathbb{C}h_0)_f = \mathbb{C}_{f,.} h_0 = 0$. This automatically implies that the covariance of $(\mathbb{C}h_0)_f$ with any other random variable is also zero. Therefore, we only need to prove that $Cov\left((\mathbb{C}h_0)_{f_1}, (\mathbb{C}h_0)_{f_2}\right) = 0$ when $f_1, f_2$ both are in $\mathcal{F}_{out}$. Note that:

$$(\mathbb{C}h_0)_{f_1} = \sum_{f \in \mathcal{F}} \mathbb{C}_{f_1,f} h_{0,f} \quad \text{and} \quad (\mathbb{C}h_0)_{f_2} = \sum_{f \in \mathcal{F}} \mathbb{C}_{f_2,f} h_{0,f}.$$

Therefore,

$$Cov\left((\mathbb{C}h_0)_{f_1}, (\mathbb{C}h_0)_{f_2}\right) = Cov\left(\sum_{f \in \mathcal{F}} \mathbb{C}_{f_1,f} h_{0,f}, \sum_{f \in \mathcal{F}} \mathbb{C}_{f_2,f} h_{0,f}\right)$$

$$= \sum_{f \in \mathcal{F}} \sum_{f' \in \mathcal{F}} (h_{0,f} \cdot h_{0,f'}) \cdot Cov(\mathbb{C}_{f_1,f}, \mathbb{C}_{f_2,f'})$$

We now argue case by case:

- $f, f' \in \mathcal{F}_{out}$: In this case, $Cov(\mathbb{C}_{f_1,f}, \mathbb{C}_{f_2,f'}) = 0$ because there can be no edges from either $f_1$ or $f_2$ to $f$ or $f'$ since all of them are nodes in $\mathcal{F}_{out}$.

- $f \in \mathcal{F}_{out}, f' \in \mathcal{F}_{in}$: In this case, $\mathbb{C}_{f_1,f} = 0$ by the same argument as above. Therefore, the covariance must be 0.

- $f' \in \mathcal{F}_{out}, f \in \mathcal{F}_{in}$: In this case, $\mathbb{C}_{f_2,f'} = 0$ which makes the covariance 0.

- $f \in \mathcal{F}_{in}, f' \in \mathcal{F}_{in}$: Finally, if both $f$ and $f'$ are in $\mathcal{F}_{in}$, there can be edges from $f_1$ and $f_2$ towards $f$ and $f'$. But those edges are disjoint and therefore, independent which makes the covariance term 0.

This concludes the proof. $\qquad\square$

# E Supplementary Proofs

## E.1 Computation of the Contribution Matrix $\mathbb{C}$

**Proposition 9.** *Given adjacency matrix $A$, the contribution matrix of causal graph $\mathcal{G}$ is given by*

$$\mathbb{C} = \sum_{k=0}^{|\mathcal{F}|} A^k,$$

*and therefore can be computed in polynomial time in $|\mathcal{F}|$.*

*Proof.* The key step to complete the proof is to show that $A_{ij}^k$ captures the influence exerted by feature $i$ on feature $j$ through a directed path on the graph that is exactly $k$ hops long. We will prove by induction.

**Base case ($k = 0$):** When $k = 0$, there exists no directed path from feature $i$ to feature $j$ unless $i = j$. Therefore, all off-diagonal entries are $0$. The only entries appear on the diagonal because feature $i$ affects itself with a unit positive multiplier. This gives us the identity matrix in $|\mathcal{F}|$ dimensions which is exactly given by $A^0$.

**General case:** Suppose that the induction hypothesis holds for some $k > 1$. We will now show that it also holds for $k + 1$. Note that:

$$A_{ij}^{k+1} = \sum_{n=1}^{|\mathcal{F}|} A_{in}^k \cdot A_{nj}.$$

Since the induction hypothesis is true, $A_{in}^k$ captures the influence exerted by feature $i$ on feature $n$ through a directed path exactly $k$ hops long. $A_{nj}$ represents the direct influence exerted by feature $n$ on feature $j$ (in exactly 1 hop). Therefore, the product measures the influence of feature $i$ on feature $j$ exerted on a directed path $k + 1$ hops long. The sum over all features in $\mathcal{F}$ captures all such directed paths from $i$ to $j$. Thus, our induction hypothesis is also true for $k + 1$.

Finally, to compute $\mathbb{C}$, we need to sum the influences of directed paths of all lengths starting at node $i$ and ending in node $j$. Since $\mathcal{G}$ is a directed acyclic graph with $|\mathcal{F}|$ nodes, the length of the maximum directed path from $i$ to $j$ is at most $|\mathcal{F}| - 1$ hops long or conservatively $|\mathcal{F}|$ hops long (note that if there are no directed paths of length $k$ from $i$ to $j$, $A_{ij}^k = 0$. So, it does not hurt to be conservative). This leads to the final expression of $\mathbb{C}$:

$$\mathbb{C} = \sum_{k=0}^{|\mathcal{F}|} A^k.$$

To conclude the proof, we need to argue about the time complexity of computing $\mathbb{C}$, given matrix $A$. Multiplying 2 matrices of size $|\mathcal{F}| \times |\mathcal{F}|$ takes $O(|\mathcal{F}|^3)$ time and we need to execute $O(|\mathcal{F}|)$ such matrix multiplication steps to compute the different powers of $A$. Therefore, the overall time complexity is polynomial in $|\mathcal{F}|$. $\square$

## E.2 Proof of Supporting Result in Proposition 3

We made the following observation in our proof of Proposition 3:

**Observation.** *Let $X \in \mathbb{R}^n$ and $M \in \mathbb{R}^{n \times n}$. Define set $Y$ as follows:*

$$Y := \{y : \exists\, x \in X \quad s.t. \quad y = Mx\}$$

*When $M$ is full row-rank, set $X$ is convex if and only if set $Y$ is convex.*

We provide a formal proof here. We need to show both directions.

($\Longrightarrow$) Suppose, set $X$ is convex. We need to show that set $Y$ is convex. Let $y_1, y_2 \in Y$ such that $y_1 \neq y_2$. Pick any $\lambda \in [0, 1]$. Then there must exist $x_1, x_2 \in X$ such that $y_1 = Mx_1$ and $y_2 = Mx_2$. Clearly $x_1 \neq x_2$. Since $X$ is a convex set, $\lambda x_1 + (1 - \lambda)x_2 \in X$. This implies,

$$\lambda y_1 + (1 - \lambda)y_2 = \lambda Mx_1 + (1 - \lambda)Mx_2$$
$$= M\left(\lambda x_1 + (1 - \lambda)x_2\right) \in Y.$$

( $\Longleftarrow$ ) For the other direction, we assume that set $Y$ is convex and we need to show that set $X$ is convex. Pick any two elements $x_1, x_2 \in X, x_1 \neq x_2$ and any $\lambda \in [0, 1]$. Let $y_1 = Mx_1$ and $y_2 = Mx_2$. Clearly, $y_1, y_2 \in Y$ (by definition). Note that $y_1 \neq y_2$ (otherwise, we would have $Mx_1 = Mx_2$ which implies that $x_1 - x_2 \in \text{Nullspace}(M)$. But $\text{Nullspace}(M) = \emptyset$ as $M$ is full row-rank). Additionally, $\lambda y_1 + (1 - \lambda)y_2 \in Y$ since $Y$ is a convex set. This implies,

$$\lambda x_1 + (1 - \lambda)x_2 = \lambda M^{-1}y_1 + (1 - \lambda)M^{-1}y_2 \quad (M^{-1} \text{ exists because } rank(M) = n)$$
$$= M^{-1}\left(\lambda y_1 + (1 - \lambda)y_2\right) \in X.$$

The last part follows from noting that $\lambda y_1 + (1 - \lambda)y_2 \in Y$ and since $M$ is full row-rank, the pre-image of $\lambda y_1 + (1 - \lambda)y_2$ must be unique. This concludes both directions of the proof.

