# OpenReview forum: "Incentivizing Desirable Effort Profiles in Strategic Classification: The Role of Causality and Uncertainty"
_NeurIPS.cc/2025/Conference — NeurIPS 2025 poster_

### Official Review · Reviewer_RF8Z · 2025-06-07

**Clarity:** 3
**Significance:** 2
**Originality:** 2
**Rating:** 4
**Confidence:** 3

**Summary:**

This paper proposes a new formulation of SC based on a causal perspective. Specifically, the causal effects of investing efforts are captured by a contribution matrix $C$ and the decision maker publishes a linear classifier $h$. The cost is formulated as the weighted $L_p$ norm of the effort. The paper first characterized the optimal effort profiles when agents best respond under a complete information setting, and pointed out that space of desired classifiers is non-convex/convex when the set of desirable features has cardinality $>1$/$1$. Finally, the paper discussed two settings of incomplete information and gave explicit expressions of the optimal effort profile under when the cost is $l_2$.

**Questions:**

1. Are there efficient algorithms to learn an optimal classifier to incentivize desirable efforts in your problem setting, especially when there is incomplete information?
2. Can you discuss with learning the contribution matrix $C$ in practice?

**Ethical Concerns:**

["NO or VERY MINOR ethics concerns only"]

**Final Justification:**

- Thanks for pointing me to Appendix A. The discussion of this work and [23] makes sense. I still think his work has similar settings to [23], but it is slightly more general, and this extension may be non-trivial.

- For the related work, I did not mean to say your work is similar to those, but just hope most works closely related are referred.

- I encourage the authors to add more discussions on algorithm design.

Overall, I think the paper has merit and will retain the positive score.

**Limitations:**

yes

**Quality:**

3

**Strengths And Weaknesses:**

> Strengths
- The problem formulation is reasonable and the theoretical results seem to be solid.
- The consideration of incomplete information is interesting.
- The writing is clear.

> Weaknesses
- The formulation of the causal structure is still similar to [23] although [23] did not mention "causal". The causal effects are still characterized by the multiplication of $C$ and $e$. But the causal effects in reality can be more complex (e.g., non-linear)
- Getting the contribution matrix $C$ may be difficult in practice, so it may be worthwhile adding such a discussion on the practical usage.
- Some highly related works are not mentioned. For example, [1] discussed long-term genuine improvement in SC settings, while [2] discussed the externalities of SC.

[1] Xie, T., Tan, X., & Zhang, X. (2024, October). Algorithmic decision-making under agents with persistent improvement. In Proceedings of the AAAI/ACM Conference on AI, Ethics, and Society (Vol. 7, pp. 1672-1683).

[2] Hossain S, Micha E, Chen Y, et al. Strategic classification with externalities[J]. arXiv preprint arXiv:2410.08032, 2024.

---

> ### Author Rebuttal · Authors · 2025-07-30
>
> We thank the reviewer for the careful read of our paper, their positive feedback about the strengths of our work, and their insightful comments. Below, we reply to the weaknesses and questions noted by the reviewer:
>
> *Weaknesses*
>
> W1. We want to highlight that much of the strategic classification literature that models causal effects only considers linear causal relationships (which is the same approach we adopt here) (Kleinberg and Raghavan (2020), Shavit et al (2020)). Our main goal in this work is to take a first step towards studying the problem of incentivizing desirable agent behavior when agents have uncertainty not only in the classifier, but also in the causal graph. Linear causal graphs are the first natural point of attack for understanding the effects of causality and incentivizing certain types of behavior in strategic classification; the reason being that when it comes time to generalize to non-linear settings, we will be able to disentangle the effects of (non)-linearity in the causal structure from the conclusions. Additionally, we have already seen that the problem is already non-trivial under the linearity assumption, which hints at the fact that it is likely to be intractable under general non-linear causal structures. However, we do agree with the reviewer that there are real-world instances where causal relationships are complex/non-linear. Considering non-linear causal relationships could be an exciting avenue for future research.
>
> We agree that Kleinberg and Raghavan [23] also consider causal relationships between features. We have highlighted in Appendix A (Related Work) that our work differs from [23] in the following main aspects:
>
> a) We consider more general cost functions and show that the nature of the cost function significantly affects the best response effort profile and consequently, the principal’s problem of designing desirable classifiers.
>
> b) We consider incomplete information where the agent can have uncertainty in both the classifier and causal graph. [23] does not consider uncertainty at all.
>
> c) In our work, we introduce a general and flexible notion of desirability of effort parametrized by $\beta$: we say a profile is desirable if the total effort on desirable features is at least a $\beta$-fraction of the total effort. This is in sharp contrast to [23] where desirability of an effort profile is pre-determined by the principal.
>
> W2. Please refer to the response to Q2 below.
>
> W3. We provide a summary of how our work differs from the two works referenced by the reviewer:
>
> While the high-level setting introduced in Xie et al (2024) is also about improvement of agents, we highlight some key differences: i) firstly, their main goal is to understand the temporal aspects of agent behavior (where change in behavior materializes slowly over a long period of time), while we try to understand how agents best-respond to a classifier under incomplete information—our settings and goals are therefore orthogonal—, ii) our work considers causality in features, introducing inter-dependencies across features, but Xie et al (2024) operates in a simpler independent feature setting, iii) in their work, the principal has a fixed desirable vector ‘d’ and all agent behaviors are measured with respect to d (alignment <q,d>)---this is in sharp contrast to our work where we introduce a general, more flexible concept of beta-desirability that can evaluate a wider range of human behavior more meaningfully, iv) finally, because of their simple evaluation rule, Xie et al (2024) have a single-variable decision problem for the principal (designing the threshold value that separates gaming from improvement). However, our work captures most real-world strategic classification settings where the principal actually tries to design a high-dimensional classifier in order to incentivize “desirable behavior”. We can add a short discussion about this in the revised version of the paper.
>
> The main focus of Hossain et. al (2024) is fundamentally different from ours, as it centers on externalities arising from interdependence between agents. Beyond the difference in the main question, their tools and modeling are completely different—there is no notion of causality or improvement, and they use game-theoretic modeling and solution concepts. We are happy to add a discussion of this paper.
>
>
> *Questions*
>
> Q1. Our main contribution in this paper is to understand how easy/difficult the principal’s problem of incentivizing desirable agent behavior is, when the agent has different levels of access to information; in some sense, we take an information theoretic point-of-view, rather than a prescriptive one.
>
> Since the problem is generally non-convex, we have thought about efficient algorithm design by providing some heuristics (see e.g., Appendix C.4). We find it useful to highlight some key aspects of efficient algorithm design in this setting:
>
> i) first, desirability does not come for free; in particular, there is a trade-off between \beta-desirability and accuracy. So, the first step for algorithm design is to characterize the pareto-front of this trade-off. This would provide us with a baseline (best achievable accuracy under $\beta$-desirability, given $\beta$).
>
> ii) We have seen that even under partial uncertainty, characterizing agent effort profiles in closed form is difficult. So, the intuition is to use the structural property of the optimal effort profile (Thm 5 - agents under uncertainty invest effort in features with high expected importance and low uncertainty) to design good approximation algorithms for finding the optimal classifier.
>
> iii) Finally, greedy algorithms do seem to be a natural choice for the principal when it comes to classifier design. We will add this discussion on Section 6 Discussion.
>
>
> Q2. Thanks for this very insightful question. There are several ways an agent can possibly learn the environment. One way could be through peer-learning—agents may interact with other agents which can help them learn $C$ over time—, similarly to (Bechavod et al. 2022). From a technical point of view, note that agents do not need to learn $C$ directly—all their decisions depend only on the vector $Ch$. In turn, the agent learning problem is that of learning a $d$-dimensional vector. Under i.i.d., noisy samples of effort profiles and resulting features from peers, this is a well-understood problem that follows standard concentration inequalities—we did not include those to keep the paper to the point. Under non-noisy, perfect samples, we can learn $Ch$ fully. These are great questions and will be very important for future work.

---

> > ### Comment · Reviewer_RF8Z · 2025-07-31
> > **Thanks for the rebuttal**
> >
> > - Thanks for pointing me to Appendix A. The discussion of this work and [23] makes sense. I still think his work has similar settings to [23], but it is slightly more general, and this extension may be non-trivial.
> >
> > - For the related work, I did not mean to say your work is similar to those, but just hope most works closely related are referred.
> >
> > - I encourage the authors to add more discussions on algorithm design.
> >
> > Overall, I think the paper has merit and will retain the positive score.

---

### Official Review · Reviewer_sN5p · 2025-06-23

**Clarity:** 3
**Significance:** 3
**Originality:** 3
**Rating:** 3
**Confidence:** 4

**Summary:**

The paper proposes a causal framework for strategic classification in which agents invest effort to modify features that influence others, and a principal wants to steer that effort toward desirable attributes. It introduces $\beta$-desirability to quantify the share of effort on preferred features, characterizes agents’ optimal effort under $\ell_p$ costs, and derives sufficient conditions on linear classifiers that guarantee desirable behavior under full information. The authors show finding such classifiers is generally non-convex but becomes convex when one feature is targeted, and they analyze partial-information settings, providing closed-form responses and numerical experiments on cardiovascular-risk data to validate their theory.

**Questions:**

- Your model presumes agents know the entire causal-graph topology. How would the guarantees change if this topology were uncertain or misspecified?

- How does the proposed method scale as the number of features $d$ grows large?

- The empirical study uses only one eight-feature CVD graph. Can you provide additional experiments on public strategic-behaviour datasets (e.g., credit scoring, résumé screening) to demonstrate external validity?

**Ethical Concerns:**

["NO or VERY MINOR ethics concerns only"]

**Final Justification:**

**Clarified in rebuttal:**

* **Graph knowledge:** Agents need beliefs over **$Ch$** (importance vector), not full DAG; main insights persist.
* **$\beta$ feasibility:** Achievable $\beta$ is **instance-dependent**; as $\beta \to 1$ feasibility shrinks (clear example provided).
* **Assumptions:** Will explicitly justify linear models/Gaussian priors and explain why broader regimes are intractable.
* **Recourse link & clarity:** Added positioning vs. recourse; plan to streamline theorem statements and add intuition.

**Key limitations (remain):**

* **Empirics:** Single 8-feature CVD DAG; no cross-domain baselines or stress tests.
* **Design guarantees:** Principal’s problem is **non-convex**; only heuristics, no performance guarantees.
* **Scope:** Linear $C$, linear classifiers, Gaussian beliefs; limited guidance on scalability/misspecification.


The primary remaining concern is empirical evaluation. The authors argue that the CVD study is the only suitable dataset, but they do not assess their method on standard synthetic benchmarks or public datasets with ground-truth structural equations commonly used in the causal community. During the rebuttal, they chose not to add such tests. This materially weakens the empirical case and, accordingly, lowers my overall recommendation.

**Limitations:**

Yes

**Quality:**

3

**Strengths And Weaknesses:**

## Strengths

* Well-structured paper.  Clear separation of complete vs. incomplete information.

* Combines causal modelling with strategic classification under uncertainty—areas that had largely been studied separately.

* Good intuitive explanations. After each theorem the authors add discussion.

* All main results are formally stated with complete assumptions and full proofs in the appendix.


## Weaknesses

* **Causal Structure**:  The entire DAG topology is assumed common knowledge; in practice, agents seldom know the full graph structure.

* **Feature Dynamics**: The framework relies on linear causal relations encoded in a contribution matrix $C$, whereas real-world effects are often non-linear or state-dependent.

* **Finite Sample**: The paper does not address how estimation error in the DAG would affect the results.

* **$\beta$-Desirable**: just only large amount of $\beta$-desirable provied gurantee to use desirable faetures instead of undesireable feature

* **Classifier Family**: Analysis is limited to linear classifiers with a fixed threshold, while non-linear models (trees, neural nets) dominate many real applications.

* **Agent information**: Agents’ beliefs over $h$ and edge weights are assumed Gaussian, yet empirical priors are often multi-modal or heavy-tailed.

* **Non-convex design problem.** The desirable-classifier search is non-convex except when $|D| = 1$ and $p \le 3$; the paper provides only a heuristic with no guarantees in the general case.

* **Agent optimisation hardness.** Under full uncertainty the best-response program is non-convex (they show even the simple 2-feature toy becomes non-concave) , but no algorithmic remedy is proposed.

* **Algorithmic Recourse**: Despite close ties, the paper does not reference the algorithmic recourse literature.

* **Simulation**: The only experiment uses one expert-elicited CVD DAG with eight features; no cross-domain validation, no real behavioural data, and no statistical uncertainty.

---

> ### Author Rebuttal · Authors · 2025-07-30
>
> We thank the reviewer for the careful read of our paper, their positive feedback about the strengths of our work, and their insightful comments. Below, we reply to the weaknesses and reviewers’ questions:
>
> *Weaknesses*
>
> W1. Kindly refer to Q1 below.
>
> W2. We highlight that much of the strategic classification literature that models causal effects only considers linear causal relationships (which is the same approach we adopt here) (Kleinberg and Raghavan (2020), Shavit et al (2020)). Our main goal in this work is to take a first step towards studying the problem of incentivizing desirable agent behavior when agents have uncertainty not only in the classifier, but also in the causal graph. We have already seen that the problem is already non-trivial under the linearity assumption, which hints at the fact that it is likely to be intractable under general non-linear causal structures.
> We do agree with the reviewer that there are real-world instances where causal relationships are non-linear. This is an exciting avenue of future research. Another avenue is to understand how more general causal graphs can be linearized and at what cost to accuracy. We will include this short discussion in Section 6.
>
> W3. There are several ways an agent can possibly learn the environment. One is peer-learning—agents may interact with other agents which can help them learn C over time, similarly to (Bechavod et al. 2022). From a technical point of view, agents do not need to learn $C$ directly—all their decisions depend only on the vector $Ch$ - the agent learning problem is that of learning a $n$-dimensional vector. Under i.i.d., noisy samples of effort profiles and resulting features from peers, this is well-understood and follows standard concentration inequalities—we did not include those to keep the paper to the point. Under non-noisy, perfect samples, we can learn $Ch$ fully.
>
> W4. We are not exactly sure we understand the statement. We would be happy to address it during the discussion if the reviewer could clarify their comment.
>
> W5. Linear classifiers are commonplace across many real-world high-stakes applications due to their *simplicity*, *high degree of interpretability*, *low training costs* and *robustness to overfitting*. Ex: linear models continue to be used extensively in domains like healthcare (for disease-risk prediction and resource allocation), insurance (for predicting likelihood of claim filing or insurance fraud), and economics. For this reason, strategic classification has primarily been studied in literature on linear/binary classifiers (and we adopt the same approach). Inducing desirable behavior is also relevant in any real-world context where humans are “algorithmically scored” and there is scope for strategic behavior—see Kleinberg and Raghavan (2020) for additional, detailed examples of such settings.
>
> W6. While unimodal priors do not necessarily arise in *every* setting, they are still natural and relevant to many. In bank loans or credit scoring applications, agents typically have a sense of which features are relevant, but may have some uncertainty exactly on how much each feature matters. E.g., agents understand that lowering credit usage or re-paying loans most consistently will improve their score. Such settings are typically well approximated by unimodal distributions. Beyond strategic classification, Gaussian priors are used commonly in the ML, CS, and economics literature to model incomplete information for agents (see Elzayn et al at EC 2019, Kong et al at AAAI 2020, Acharya et al. at SATML 2023 to only name a few examples).
>
> Second, we want to refer the reviewer to Jagadeesan et. al (2021) which establishes alternative microfoundations for strategic classification that also uses a Gaussian noise model. Other recent papers in the context of strategic classification have also used similar Gaussian assumptions (e.g., Braveman and Garg at FORC 2020, Avasarala et al. at AIES 2025).
>
> W7. Our main contribution is to understand how easy/difficult the principal’s problem of incentivizing desirable agent behavior is, when the agent has different levels of access to information. Providing algorithms with provable guarantees is a key future direction that we are actively pursuing.
> We want to highlight some key aspects of classifier design under uncertainty. We have seen that even under partial uncertainty, characterizing agent effort profiles in closed form is difficult. So, the intuition is to use the structural property of the optimal effort profile (Thm 5 - agents under uncertainty invest effort in features with high expected importance and low uncertainty) to design good approximation algorithms for finding the optimal classifier. Finally, greedy algorithms that pick and optimize over one desirable feature at a time do seem to be a natural choice for the principal.
>
> W8. Under general uncertainty (incomplete info over classifier + causal graph), we show that the agent problem itself is not tractable. Τhis suggests that the agent may attempt to best-respond approximately, rather than optimally. Modeling such approximate best-responses opens up new avenues for future research, particularly in understanding the behavioral dynamics and outcomes when agents operate under bounded rationality or limited information.
>
> W9. As the reviewer notes, recourse is directly related, and we will add a discussion of recourse in the related work. From a mathematical standpoint, strategic classification and algorithmic recourse can be seen as flip sides of each other: in recourse, the learner tells the agent what actions to take (but the agents may decide to take a different action unless properly incentivized) while in strategic classification, the agents decide on their own actions based on the classifier.
> We note, however, several differences between recourse and strategic classification:
>
> a) Practically, recourse requires providing a recommended action to each agent who obtains a negative decision, and is still less common in practice than releasing a single, often not fully transparent, model. Strategic classification, with incomplete information, allows us to capture such practical scenarios.
>
> b) The addition of the beta-desirability is novel. As we show, this constraint makes the problem significantly more difficult, leading to non-convexities. We expect these issues to also arise when adding desirability to the algorithmic recourse literature, since the optimization problems solved by both fields are similar, and we think that the inclusion of desirability in the recourse literature is an exciting direction for future work.
>
> c) Much of the recourse literature aims to find a low-cost path between an agent's current features and features that lead to positive classification. Importantly, much of the literature does not think specifically about whether that path involves gaming or modifying undesirable features. König et al. "Improvement-focused causal recourse (ICR)."’s paper at AAAI 2023 show that standard counterfactual recourse algorithms often lead to undesirable outcomes such as gaming the system, an important limitation: as agents game the classifier, the classifier loses in accuracy and robustness, and must be modified to compensate for this accuracy loss, reducing the effectiveness of the recourse. This includes causal algorithmic recourse, such as [Karimi et al. 2021], where the causality is typically used to ensure accuracy of the recourse by taking into account how different features affect each other, whereas previous work that effectively assumed independence of features and could mis-estimate how proposed recourse would translate into feature changes. In much of this literature, however, causality is not introduced to prevent gaming vs improvement or understand desirability.
>
>
> W10. Kindly refer to the Q3 below.
>
>
> *Questions*
>
> Q1. Recall, we show in Theorem 4 that the agent’s optimization problem under uncertainty is tractable when their priors over the importance vector $Ch$ is gaussian — an important contribution of our paper involves microfounding this using more primitive models of incomplete information (priors over the classifier and the graph weights while fully knowing the topology). However, note that the most crucial piece of information that agents need in order to make decisions is the overall importance vector of features, given by $Ch$ (it subsumes information about both the classifier and causal graph). Therefore, our main theoretical insights would go through even absent the assumption about fully knowing the graph topology, as long as the agent has some partial information about the overall importance of individual features.
>
> Q2. As the number of features $d$ grows, the agent’s cognitive load for best-responding increases (as they now have to consider much larger contribution matrices). We touch upon this aspect in detail in Section 6 Discussion. However, all of our theoretical results go through irrespective of the value of $d$. Note that all non-convexity results remain; the size of all vectors used in the desirability constraint hence the complexity of the problem in the convex case scale linearly in $d$.
>
> Q3. We want to highlight that there are some challenges with the publicly available datasets highlighted by the reviewer. Note that a key requirement for conducting numerical experiments in our setting is the availability of the corresponding causal graph. However, the data contained in these public datasets is purely observational and does not contain interventions or counterfactuals which can be used to recover the true causal graph. The issue with using such data is that causal graphs cannot be point identified and resulting graphs are often severely misaligned with reality, in particular increasing the risk of societal harm due to misinformed decisions. The primary reason we use the CVD-risk study is that the causal graph for that study is publicly available.

---

> > ### Comment · Reviewer_sN5p · 2025-08-01
> >
> > Thank you for the detailed rebuttal. Unfortunately, the key concerns remain largely unresolved. The paper continues to assume agents know the entire causal graph, confines itself to linear causal relations, linear classifiers, and Gaussian priors, and addresses the principal’s non-convex design only with heuristics that carry no performance guarantees. Your explanation for avoiding additional public datasets—that suitable causal graphs are unavailable—overlooks common practice: many empirical studies elicit, infer, or simulate causal structure from publicly released data, and synthetic benchmarks are routinely used to stress-test robustness. Because the method is evaluated only on a single, eight-feature CVD graph and never against baseline approaches, its contribution is primarily mathematical rather than actionable. Finally, my request in W4 for a concrete clarification of the $\beta$-desirability guarantee in Theorem 2 remains unanswered.

---

> > > ### Author Response · Authors · 2025-08-02
> > > **Author Response**
> > >
> > > Thank you for responding to our rebuttal and for engaging further with us! We think that there are some misunderstandings that we want to clarify:
> > >
> > >
> > > 1. Quoting your comment on W4 verbatim here "\beta-Desirable: just only large amount of \beta-desirable provied gurantee to use desirable faetures instead of undesireable feature" – we cannot understand what you mean in this comment. Quoting from our rebuttal: “W4. We are not exactly sure we understand the statement. We would be happy to address it during the discussion if the reviewer could clarify their comment.” **We would like to re-iterate the request for clarification**, especially since this seems to be an important part of the reviewer’s evaluation criteria.
> > >
> > >
> > > 2. In our rebuttal, we have provided **detailed** justification for all of our assumptions including linear classifiers (W5), linear causal graphs (W2) and Gaussian priors (W6), alluding to several real-world scenarios where these assumptions hold. We have also provided references to multiple papers in top venues that have similar assumptions. There is always a trade-off between how complicated a model can be and how analytically tractable it is. *Our main contributions are theoretical, indeed, but we want to highlight that this is a significant leap from what is the state of the art in terms of theory in strategic classification*; as such, they are the frontier of what we know for human incentives in algorithmic decision-making.
> > >
> > >
> > > 3. We have also clarified (in response to Q1) that the assumption of the agent knowing the causal graph topology is **not** strictly necessary. **All our main insights go through even if the agent has some partial information about the importance vector $Ch$ without having to know the causal graph topology**.
> > >
> > >
> > > 4. Regarding experiments, we first want to point out that *according to NeurIPS PC instructions, no additional experiments can be provided during rebuttal*. We also disagree with the reviewer’s comment that causal graphs can be reconstructed from publicly available data. There is a subtle difference between reconstructing any causal graph and reconstructing a causal graph that works in our setting. **In our setting, knowing that one feature affects another is not enough, we also need to know how much changing one feature can change the downstream feature. This information cannot be re-constructed without access to counter-factual data or data about interventions**, which most studies do not have. The CVD-risk case-study (which we use) is unique in the sense that it not only provides the causal relationships between features, but also consensus scores from multiple experts on each of the causal relationships. This enables a reasonable reconstruction of the causal graph. Please refer to Appendix B.1 Experimental Setup – Building the Causal Graph. One can always argue: why not  simulate the weights of the graph? Unfortunately, that makes the setup entirely synthetic and not really useful. Did the reviewer have something else in mind? Finally, we have highlighted that **the insights obtained from the CVD-risk model are qualitative, validate insights from theory, and will extend to any other general strategic classification setting**. We’re always happy to engage further about the value of empirical validation in such a theory-driven paper as ours.
> > >
> > >
> > > 5. [Heuristics for non-convex principal problem with no guarantees] We wish to highlight that the primary goal of our work was to characterize how easy/hard the principal problem of incentivizing desirable behavior is, under causality and incomplete information. We believe that we have achieved that goal with a fair amount of success, clearly characterizing the difficulty of the problem as a function of the level and source of uncertainty in the environment. The heuristic provides an initial insight about how to go about designing algorithms for this problem in the general case in the complete information setting. We have also provided a detailed discussion on our initial thoughts on the question of algorithm design finding desirable classifiers under incomplete information during the initial rebuttal (response to W7).
> > >
> > >
> > > Again, thank you for engaging with us! If there is anything else we can help to clarify (e.g. with regards to your stated W4), we are happy to do so.

---

> > > > ### Comment · Reviewer_sN5p · 2025-08-04
> > > >
> > > > 1. **Clarification on $\beta$-desirability.**:
> > > >     I apologise if my earlier remark was unclear. My understanding is that larger values of $\beta$ are considered more desirable, yet in Theorem 2 (line 197) the right-hand side grows unbounded as $\beta \to 1$. Consequently, achieving $\beta$-desirability appears to become increasingly difficult unless the sum of desirability on the left-hand side is correspondingly large. Could you elaborate on any practical upper bound for $\beta$ when a substantial share of features is undesirable?
> > > >
> > > > 2. **Role of simplifying assumptions.**
> > > >     I fully recognise and appreciate that your contribution is primarily theoretical. My point is that when we adopt strong simplifying assumptions, it is useful to articulate why more general cases remain out of reach, so that readers appreciate the value of the tractable regime you analyse.
> > > >
> > > > 3. **Partial knowledge of the causal graph.**
> > > >     You note that agents need only partial information rather than the full DAG. Could you clarify how stochastic errors in that partial knowledge affect the results? In particular, if an agent misidentifies their local structure, does the framework still hold? It would help to understand whether the assumption of knowing the full DAG and that of knowing one’s local neighbourhood differ materially in practice.
> > > >
> > > > 4. **Experiments and alternative datasets.**
> > > >     You mention that NeurIPS discourages additional experiments in the rebuttal. I could not locate that restriction; the NeurIPS FAQ ([https://neurips.cc/Conferences/2025/PaperInformation/NeurIPS-FAQ](https://neurips.cc/Conferences/2025/PaperInformation/NeurIPS-FAQ)) states:
> > > >
> > > > > **“Can the rebuttal include new results?** Yes, however, your original submission will serve as the basis …”
> > > >
> > > >  Separately, while the CVD-risk study is indeed useful, the causal-inference community offers several publicly available data sets that include either interventional data or ground-truth structural equations. These could complement your case study and demonstrate the robustness of your approach beyond a single domain.
> > > >
> > > > 5. **Balance between theory and empirics (see point 2).**
> > > >     If the work is positioned as primarily theoretical, a minimal empirical section is perfectly acceptable. My reservation arises only when empirical claims rely on one unique data set; broadening or contextualising that evidence would strengthen the overall contribution.

---

> ### Author Response · Authors · 2025-08-06
> **Author response**
>
> We thank the reviewer for their thoughtful response and for giving us a chance to answer their questions.
>
> **Q1. [Clarification on $\beta$-desirability]** The reviewer is correct; higher $\beta$ indicates a higher degree of desirability. This makes desirability harder to satisfy, reducing the size of the space of desirable classifiers. When $\beta = 1$, the requirement becomes extreme: it requires that *the entire effort* is put on desirable features, with *none on undesirable features*. This is mostly impossible—this needs the RHS of our expression to be 0, which only happens when agent has no benefit to modify any undesirable feature.
>
> A natural way to upper-bound $\beta$ is to find the limiting value of $\beta$ beyond which the desirability constraint is infeasible. This gives us an instance-specific bound—such bounds will depend on the causal graph. As mentioned above, $\beta = 1$ can be reached when$(C h)_{U} = 0$. This can be achieved, for example, when all features are independent ($C = I_d$), by setting classifier weights of all undesirable features = 0.
>
> Now, consider the following example: there are 2 features, feature 1 (undesirable) causally affects feature 2 (desirable). Suppose the edge weight is 1. In this case, we can show that when searching over classifiers $h \geq 0$, the best achievable $\beta$ is $\frac{1}{\sqrt{2}} = 0.707$ (for agents with unweighted $\ell_2$-costs). Proof: For this graph, $C = \begin{bmatrix}1 & 1 \\\ 0 & 1 \end{bmatrix}$. The desirability constraint is: $h_2 \geq \frac{\beta}{\sqrt{1-\beta^2}} (h_1 + h_2)$ where $h_1, h_2 \geq 0$. Clearly, when $ \frac{\beta}{\sqrt{1-\beta^2}} > 1 \iff \beta > \frac{1}{\sqrt{2}}$, no feasible non-trivial classifier $h$ exists.
>
> We hope that these examples drive the point that the best achievable $\beta$ depends on the graph characteristics.
>
>
> **Q2. [Role of Simplifying Assumptions]** Thank you for recognizing and appreciating that our contribution is primarily aimed at advancing the theory of strategic classification. We fully agree with your point that in a theory paper, it is important to convey the challenges that arise in a general setting so that readers understand the value of the assumptions and what it adds to the analysis in terms of tractability. We will add more concrete discussions about the assumptions in the final version. We thank you for pointing this out to improve the broad appeal and readability of our work.
>
>
> **Q3. [Partial Knowledge of Causal Graph]** There are 2 parts to this question. Firstly, our framework can seamlessly handle the scenario where there are errors in the agent’s partial information pertaining to the graph; this incorrect prior will just be replaced in optimization problem (4) (see next paragraph). The agent’s prior is meant to model their belief, which may or may not match the ground truth. None of our results depend on whether agents have a correct prior about the environment or not, so long as the principal can infer said prior.
>
> That being said, an agent having an erroneous prior might make it more difficult to elicit desirable behavior---for example, if an agent believes that investing in undesirable features is cheap and can have a significant impact on other features, there may not be a way for the learner to steer the agent away from undesirable effort. This would be seen in a reduction of the achievable beta at the specific, instance level.
>
>
> **Q4. [Experiments and Alternative datasets]** We received an email from NeurIPS on 27 July, with instructions for the discussion period. Quoting: “For authors, please refrain from submitting images in your rebuttal with any “tricks”.....For reviewers/ACs/SACs, please do not expect the authors to submit updated results with images or any type of rich media. Instead, please discuss the current merits of the paper and clarify any misunderstanding based on textual communications.”
>
> We interpreted this email to mean that submitting additional results is not advised since conveying new results often requires uploading a pdf document or using images (ways to circumvent the character limit). We were also advised against submitting external URLs for anonymity issues.
>
>
> **Q5. [Balance between theory and empirics]** We agree that cross-validating our numerical results on multiple different settings is only going to bolster our paper. One of our goals in the last response was to illustrate some of the real challenges we faced while looking to obtain datasets/causal graphs for our setting.
>
> That being said, we note that contributions in the area of strategic classification have been largely theoretical, and *one major goal of our work was to propose new frameworks and paradigms that help take the theory one step closer to reality* (by incorporating causality and uncertainty). We understand that there is still a gap between theory and practice, but we believe our contribution helps close this gap and actually sets the tone for more empirical work.

---

> > ### Comment · Reviewer_sN5p · 2025-08-07
> >
> > Thank you for the clarification regarding \$\beta\$ and the use of graph knowledge — I believe it would strengthen the paper to include these explanations directly in the text.
> >
> > Regarding the experiments, I also ran some tests during NeurIPS 2025 and reported the results as a markdown table, in compliance with all submission guidelines. Encouragingly, we arrived at a similar conclusion: **there remains a gap between theory and practice.**
> >
> > While I am not an expert in your specific subfield, in my area (causality), it is standard practice to evaluate methods on widely accepted real-world or well-designed synthetic datasets. Given your good theoretical contributions, I still feel that this practical gap persists. Strengthening the empirical validation with such datasets could help bridge this gap and improve the impact of the work.

---

> > > ### Author Response · Authors · 2025-08-07
> > >
> > > We thank the reviewer for their continued engagement with our work. We hope our responses have addressed many of the concerns raised, and we will incorporate the promised explanations and clarifications into the final version of the paper. While we acknowledge the remaining gap between theory and practice, we believe our work takes a first step toward bridging this gap. We look forward to presenting a more thorough experimental study in a future, application-driven, follow-up to this paper.

---

### Official Review · Reviewer_Mw5J · 2025-07-01

**Clarity:** 3
**Significance:** 3
**Originality:** 3
**Rating:** 5
**Confidence:** 3

**Summary:**

This paper addresses strategic classification under causal interdependencies and incomplete information. It introduces the concept of β-desirable effort profiles and analyzes how rational agents respond to deployed classifiers under different information assumptions. The authors consider both complete and incomplete knowledge settings and provide a combination of theoretical results and simulations based on cardiovascular disease prediction. The model incorporates causal graphs, agent cost functions, and effort optimization, providing insight into classifier design that incentivizes desirable (rather than merely effective) changes.

**Questions:**

1. Could the authors contrast this work more clearly with the literature on causal algorithmic recourse (e.g., [Mothilal et al., 2020], [Karimi et al., 2021])? The lack of mention is surprising given the shared setting.
2. Are there cases where the incentive structure leads to undesirable gaming behavior under partial information? A small example would be helpful.
3. The β-desirability conditions are mathematically clean — could the authors comment on the extent to which principals can reasonably verify them in practice?
4. Line 90: is it that the features are desirable/undesirable, or changes to/interventions upon those features?
5. How would the framework handle heterogeneous populations (e.g., different groups with different priors or cost sensitivities)?

**Ethical Concerns:**

["NO or VERY MINOR ethics concerns only"]

**Final Justification:**

The rebuttal satisfactorily addressed key concerns, including clarifying modeling assumptions, the role of desirability, and connections to related work. While some empirical limitations and assumptions (e.g., linearity, Gaussian priors) remain, they are clearly acknowledged and framed as future directions. The paper makes a strong theoretical contribution to strategic ML under causal and informational constraints, and I believe its strengths justify a positive recommendation.

**Limitations:**

No — while section 6 does speak to the methodology's limitations, more explicit discussing regarding the practicality of assuming full causal graph knowledge and cost function specifications, as well as risks tied to fairness, misspecification, and strategic disparity would be welcome.

**Quality:**

3

**Strengths And Weaknesses:**

### Strengths
- The problem is important and well-motivated: incentivizing desirable feature change under strategic manipulation is a key issue in responsible ML.
- Strong theoretical depth with clean characterizations (e.g., β-desirability conditions).
- Good mix of theory and simulation; the cardiovascular disease application is relevant and realistic.
- Reasonable modeling assumptions around cost functions, agent behavior, and effort optimization.
- Good literature coverage (in Appendix) but also in main body when contrasting against most similar work.

### Weaknesses
- At times, the paper is dense and difficult to read — many sections could benefit from simplification or intuitive summarization of key ideas. E.g., several concepts (e.g., β-desirability, agent best response, contribution matrix) take effort to parse and might deter less technical readers.
- While the related work is thorough, it lacks direct contrast with causal algorithmic recourse literature — a striking omission given the similarity in goals. Esp. since those works operate on models beyond the linear variants studied here
- The empirical section is clean, but could benefit from a broader discussion of fairness and real-world risks.

---

> ### Author Rebuttal · Authors · 2025-07-30
>
> We thank the reviewer for the careful read of our paper, their positive feedback about the strengths of our work, and their insightful comments.
>
> We also apologize if parts of the paper feel difficult to read. We will provide intuitive explanations/visuals for some of the key concepts (like contribution matrix and desirability definitions); we already have these visuals in the public, extended version of our paper, but cannot add them to the rebuttal because of the formatting restrictions. We will elaborate upon the fairness aspects of our work in Section 6 (see Q5).
>
> Below we provide answers to the reviewer’s questions:
>
>
> Q1. Thank you for pointing this out! First, recourse is indeed directly related, and we will add a discussion of recourse in the related work. From a mathematical standpoint, strategic classification and algorithmic recourse can be seen as flip sides of each other: in recourse, the learner tells the agent what actions to take (but the agents may decide to take a different action unless properly incentivized) while in strategic classification, the agents decide on their own actions based on the classifier.
> We note, however, several differences between recourse and strategic classification:
>
> a) From a practical point of view, recourse requires providing a recommended action to each agent who obtains a negative decision, and is still less common in practice than releasing a single, often not fully transparent, model. Strategic classification, especially with incomplete information, allows us to capture such practical scenarios.
>
> b) The addition of the $\beta$-desirability constraint is novel. As we see in many places throughout the paper, this constraint makes the problem significantly more difficult, leading to non-convexities. We expect these issues to also arise when adding desirability to the algorithmic recourse literature, since the optimization problems solved by both fields are similar, and we think that the inclusion of desirability in the recourse literature is an exciting direction for future work.
>
> c) To the above point, much of the recourse literature aims to find a low-cost path between an agent's current features and features that lead to positive classification. Importantly, much of the literature does not think specifically about whether that path involves gaming or modifying undesirable features. [KFG 2023] in fact show that standard counterfactual recourse algorithms often lead to undesirable outcomes such as gaming the system. This is an important limitation: as agents game the classifier, the classifier loses in accuracy and robustness, and must be modified to compensate for this accuracy loss, reducing the effectiveness of the recourse. This includes \emph{causal} algorithmic recourse, such as [Karimi et al. 2021], where the causality is typically used to ensure accuracy of the recourse by taking into account how different features affect each other, whereas previous work that effectively assumed independence of features and could mis-estimate how proposed recourse would translate into feature changes. In much of this literature, however, causality is not introduced to prevent gaming vs improvement or understand desirability.
>
>
> [KFG 2023] König, Gunnar, Timo Freiesleben, and Moritz Grosse-Wentrup. "Improvement-focused causal recourse (ICR)." In Proceedings of the AAAI Conference on Artificial Intelligence, vol. 37, no. 10, pp. 11847-11855. 2023.
>
>
> Q2. Yes, there can indeed be scenarios with partial uncertainty where agents can have incentives to engage in undesirable behavior. In general, if undesirable features are not costly/easy to modify, and if there is a belief that such undesirable features may be prevalent in the classifier, then agents may prefer modifying these undesirable features. Note that this is a realistic assumption in real systems: credit scores rely significantly on credit usage, but credit usage can be improved by opening dummy credit card accounts without lowering actual spending or improving re-payments. In turn in some cases, it may not be possible to induce $\beta$-desirability due to the incomplete information; the probability with which such an event occurs depends on the prior (see Proposition 6 in the Appendix).
>
> We are not sure if that is exactly what the reviewer had in mind for this question, but we are very interested in engaging further during the discussion period.
>
>
> Q3. We thank the reviewer for appreciating the definition of $\beta$-desirability! We note that the $\beta$-desirability condition is on the agent’s exogenous effort profile $e$. Unfortunately, the principal does not observe $e$ directly, they only observe the change in feature values $\Delta x(e)$ because of effort $e$. Assuming the principal has complete information about the causal graph, they can infer $e$ by observing $\Delta x$ only when the contribution matrix $C$ is full rank (and hence invertible). But, in general, $C$ is not full rank because many different effort profiles can amount to the same final feature vector — this implies that in most cases, the principal cannot verify the beta-desirability condition.
>
>
> However, in our setting, the principal need not directly verify $\beta$-desirability, so long as the agents are rational. Our classifiers incentivize desirable behavior. This can be seen as a parallel to incentive-compatibility in classical mechanism design where specific actions can be incentivized without seeing an agent's type.
>
> This is a useful point to get across to the reader, and we will add this clarification in Section 6.
>
> Q4. The reviewer is absolutely right that we should have clarified the language in Line 90. What we mean is that investing effort to modify those features directly is desirable or undesirable. We actually clarify this in lines 91-96 through the health example. We thank the reviewer for the insightful observation.
>
> Q5. Our framework is indeed capable of handling heterogeneous populations with different costs/priors. The analysis of the agent problems for different groups goes through unchanged. The principal problem changes in the sense that the goal is now to pick a classifier that satisfies $\beta$-desirability simultaneously for all groups (we will have $k$ desirability constraints if there are $k$ different groups). However, note that all of our main insights still go through unchanged. For example, if the problem is hard to solve with one constraint, it is definitely harder with multiple constraints. Also, if the feasible space with one group is convex (like, when $|D| = 1$ and $p<= 3$), it will still be convex with multiple groups (since all the group-wise constraints are individually convex and intersection of convex sets is convex).
>
> Having multiple groups with different priors, however, has major fairness implications as identified by Bechavod et al (2022). When groups have different priors or different abilities to strategize, this disparately affects their ability to best respond (and in our setting, obtain $\beta$-desirable effort profiles). We already highlighted the heterogeneous population setting as an exciting future direction, but we will add the above discussion in Section 6 too.

---

> > ### Comment · Reviewer_Mw5J · 2025-08-05
> >
> > Thank you for your thoughtful rebuttal and clarifications. Allow me to emphasize the value of including explicit working examples—even if placed in the appendix but clearly referenced in the main text—especially in relation to Q2 (gaming under partial info), Q3 (verifiability of β-desirability), and Q5 (heterogeneous populations). These will not only help _familiar_ readers concretely trace the technical contributions, but also allow _new_ readers to better understand and build upon the ideas. Regarding Q1, the rebuttal discussion on the relationship with algorithmic recourse is appreciated. I believe the paper would benefit from a more deliberate presentation of the similarities and differences with causal recourse—perhaps even via one of the aforementioned examples—to clarify the boundary between these two lines of work.
> >
> > In any case, I appreciate the effort and contributions of this paper, maintain my positive score, and look forward to reading and sharing the final version.

---

### Official Review · Reviewer_oibv · 2025-07-03

**Clarity:** 4
**Significance:** 2
**Originality:** 2
**Rating:** 4
**Confidence:** 4

**Summary:**

The authors consider a binary classification with strategic agents. The agents can exert effort to get desired decision. The author also consider a scenario where there are desirable and undesirable features and investigate how the classifier can lead to improving desirable features.

The authors considers both complete and incomplete information and provide insight on the best response function of the agents.

**Questions:**

1. In what real world problems, the proposed model can be used?

2. How the design maker should find the optimal classifier? How should we define the optimal classifier? Because we want to have \beta-desirable outcome, we need to clearly define optimal classifier?

3. In the game theory and economics, there is notion of socially optimality. Why we cannot use this concept instead of Desirable Classfier concept proposed here?

4. How the problem can be extended to non-linear case?

5. What the agents are interdependent (effort of one impact others' utility)

**Ethical Concerns:**

["NO or VERY MINOR ethics concerns only"]

**Final Justification:**

The authors rebuttal was comprehensive and answered all the questions that I asked them. So I want to increase my score. In particular, they talked about the real world application and social optimality. Other reviewers also want to accept the paper and I agree with them.

**Limitations:**

The author have not discussed potential negative societal impact (or I could not find).

**Quality:**

2

**Strengths And Weaknesses:**

Strengths: The paper is well written. It is easy to follow and can provide some insight on how strategic classification works.

Weaknesses:

1. The scenario that they consider is very simple. While it is in the context of  machine learning, basically the authors consider a very simple two stage game between decision maker and an agent. The first order condition is used to find the best response.

2. The scenario looks very superficial. A scenario with liner classifier and know causal graph. It is not clear in what real world problem, we can apply this study.

3. In the incomplete information case, still the authors assume the agent knows the distribution over hypothesis space. This still gives a lot of information to agent leading to optimization problem 4.

4. No discussion on how the design maker should find the optimal classifier. The authors discuss that the problem is non-convex for the design maker. But no further insight or algorithm provided for solving that problem.

5. No discussion on how the results can be extended to non-linear case or a scenario where the agents are interdependent.

---

> ### Author Rebuttal · Authors · 2025-07-30
>
> We thank the reviewer for their comments. Below, we provide detailed responses to the weaknesses identified, followed by answers to the specific questions raised. We look forward to engaging further during the discussion period.
>
> *Weaknesses*
>
>
> W1. We respectfully disagree with the reviewer’s characterization that this is just a simple two-stage game between a decision-maker and an agent in the context of ML. The sequential nature of decision-making in strategic classification makes two-stage games the obvious modeling choice to understand these problems. We clarify that our contribution is not the introduction of a two-stage game model to understand strategic classification – this framework was introduced by Hardt et al (2016). Our main contribution is to study the principal’s problem of incentivizing desirable agent behavior in the context of strategic classification (following the work of Raghavan and Kleinberg (2020)), but under the real-world constraints of causality and uncertainty. We have already provided a detailed breakdown of the major differences of our work with Raghavan and Kleinberg (2020) in Appendix A (Related Work).
>
> We clarify that we do not use first order conditions to find the agent’s best response. As demonstrated in our paper, the agent’s decision problem becomes intractable once the agent does not have access to complete information. One of our main results characterizes conditions of partial uncertainty under which the agent problem is still tractable (see Thms 4 and 5 on pages 6 and 7).
>
> W2. We respectfully disagree that our scenario looks very superficial. Linear classifiers are commonplace across many real-world high-stakes applications due to their simplicity, high degree of interpretability, low training costs and robustness to overfitting; e.g., linear models continue to be used extensively in domains like healthcare (for disease-risk prediction and resource allocation), insurance (for predicting likelihood of claim filing or insurance fraud), and economics. For this specific reason, strategic classification has primarily been studied in literature on linear/binary classifiers (and we adopt the same approach). The question of inducing desirable behavior is also relevant in any real-world context where human beings are “algorithmically scored” and there is scope for strategic behavior – see also Kleinberg and Raghavan (2020) for additional, detailed examples of such settings.
>
> As for the known causal graph, we want to clarify that we assume full knowledge only in the complete information setting. In the incomplete information setting, we explicitly allow for agent uncertainty in the weights of the causal graph. The only assumption that we use is that agents always know the causal graph topology fully; this effectively means that the agents know the features and have a rough idea of where edges between features exist. However, note that the most crucial piece of information that agents need in order to make decisions is the overall importance vector of features, given by $Ch$ (it subsumes information about both the classifier and causal graph). Thus, our main theoretical insights would go through even absent the assumption about a fully known graph topology, as long as the agent has some partial information about the importance of individual features. As such, they are, to the best of our knowledge, among the most general assumptions in the strategic classification literature.
>
>
> W3. In the CS and economics literature (specifically (Bayesian) mechanism design), the standard model for agents with incomplete information is priors—probability distributions that capture an agent’s belief over unknown aspects of the environment (Myerson 1981, Kamenica and Gentzkow 2011). We emphasize that there is no distribution over classifiers or causal graphs — the classifier is chosen deterministically by the principal and the causal graph is fixed (by nature). Our prior just reflects the agent’s *belief* about what the deployed classifier and the weights of the causal graph are. Further, the assumption on having priors at the time of deciding how to strategize has been studied in the past in strategic classification (see ``Bayesian Strategic Classification’’ by Cohen et al., 2024) and published in NeurIPS.
>
> W4. We believe that the reviewer means “how to choose the desirable classifier that maximizes accuracy” when they ask “how to choose the optimal classifier”. We clarify that after showing that finding the most accurate desirable classifier is a non-convex optimization problem in the general case, we do provide 2 key ways to circumvent this issue:
>
> a) We show that by deciding to incentivize a desirable modification in exactly 1 feature, the principal can make this optimization problem convex and easily tractable (Thm 3).
>
> b) In the general case, we provide a heuristic that encourages searching for the most accurate classifier which cannot put more than a fixed amount of importance on undesirable features (Prop 5 in Appendix C.4). This again makes the search space convex and the optimization problem tractable.
>
> W5. We argued above why the linear classifier setting is important. Further, we have demonstrated throughout the paper that the problem is non-trivial even for the linear classifier setting with many open questions. Linear classifiers are the first line of attack when it comes to building theoretical models for these settings. Note also that thinking about non-linear classifiers opens a host of other questions about what types of non-linear classifiers we want to focus on, which algorithms are used to produce them etc. The extension to non-linear classifiers is a very interesting direction, but is beyond the scope of the present work.
> It is indeed interesting to study settings where agents are interdependent and some works have considered this aspect (Hossain et al 2024). However,  studying the interdependent setting meaningfully requires understanding what are the effects of uncertainty, causality, and incentivizing desirable behavior in the independent setting. Otherwise, it is unclear which conclusions are a byproduct of externalities on the agents and which conclusions are truly a byproduct of uncertainty and causality. Lastly, the independent setting is, in fact, the standard setting in strategic classification (Hardt et al. 2016, Ahmadi et al. 2021, 2022, Bechavod et al. 2021, Braverman and Garg 2020, Cohen et al. 2024, Chen et al. 2020, Dong et al. 2018, Ghalme et al. 2023, Harris et al. 2021, Hu et al. 2019, Kleinberg and Raghavan 2020, Miller et al. 2020, Milli et al. 2019, Shavit et al. 2020, Zhang et al. 2022).
>
> *Questions*
>
> Q1. We have already touched upon the aspect of real-world applications of our model. To reiterate, in any real-world setting where human beings are algorithmically scored (in the words of Kleinberg and Raghavan (2020)) and there is scope of strategic human behavior, the question of how to incentivize desirable behavior becomes critical; e.g., in the context of college admissions, a candidate can invest effort in broadly desirable activities, like taking AP classes in high school. Alternatively, they might engage in undesirable behaviors—like signing up for multiple extracurricular activities without meaningful participation in an attempt to appear more well-rounded. Incentivizing the first kind of behavior is a meaningful pursuit in this setting.
>
> Q2. In traditional strategic classification, the optimal classifier is defined as the classifier in the entire hypothesis class which minimizes classification error accounting for the agents’ ability to best-respond; this is also known as the Stackelberg equilibrium classifier. In our setting, the “optimal” classifier would be defined as the classifier which minimizes classification error among all classifiers in the hypothesis class which are guaranteed to induce $\beta$-desirable effort from rational agents. We use this exact definition in Appendix C.5  Of course, in general, depending on the specific application, the principal may adopt different objective functions to reflect their own notion of what constitutes the "optimal" classifier.
> 	Pertaining to the question of finding the “optimal“ classifier, we emphasize that we do provide insights about how to find it in the complete information setting. Kindly refer to our response to W4 above.
>
> Q3. Social optimality (aka, maximum social welfare) and $\beta$-desirability are complementary notions, and not directly exchangeable. Social optimality makes statements about maximizing the sum of the utilities of the agents and does not directly prevent gaming nor agents modifying undesirable features. Consider the following example, inspired by Kleinberg and Raghavan (2020): the learner, here a university professor, deploys a scoring rule that must decide which weight to put on homework assignments versus in-class exams. Each agent can decide to cheat or to study. Cheating has a very low cost, why studying has a significant cost. Further, the causal graph is such that an agent can only cheat on assignments, but not on in-class exams. In this case, the classifier that provides social optimality in that it maximizes agents’ utilities is the one that puts all the weight on assignments, and pushes all students to cheat. In turn, social optimality is obtained, but at the cost of the student “gaming” the system and investing undesirable effort. Arguably, in our example above, a professor may be more interested in preventing cheating than optimizing social welfare (which could be, in this example, seen as a form of grade inflation when students are cheating).
> 	An interesting direction for future work is to understand the trade-offs between these two complementary notions, one which is utilitarian (social optimality) and one that is normative (desirability). We will add this discussion in the Discussion (Section 6).
>
>
> Q4 & Q5. Please see our responses to W5 above.

---

### Note · Authors · 2025-08-12

We thank reviewers for reading our paper and engaging with us. We really appreciate your comments and suggestions. Below, we summarize our closing remarks:

1. The majority of the review team has recognized our contribution to advancing the theory of strategic classification. Our work introduces a **causal game‑theoretic framework** for strategic classification that uniquely incorporates both the **interdependencies among features via causal structure and agents' uncertainty** regarding the classifier and causal graph, pushing the frontiers of the field. We propose a general desirability notion to quantify how agents distribute effort between desirable and undesirable features, capturing a wide range of human behaviors beyond the common dichotomy of gaming vs improvement framing in prior work. Our main contribution is a complete characterization of how easy/hard the principal problem of incentivizing desirable effort is, as a function of the level of agent uncertainty, **not only in the classifier, but also the causal graph**. While not primarily aimed at algorithm design, we also touch on key aspects of designing algorithms under complete and incomplete information, and will incorporate reviewers’ suggestions in this regard. Though our work—like any theory work—makes some idealized assumptions, **it advances the modeling frontier and brings the literature closer to real-life considerations**.

2. Our contributions are primarily **theoretical**; we wish to understand what is possible in terms of incentivizing desirable behavior in settings where the agents can strategically best-respond to the classifier. We also include an empirical study on a real dataset (and a real causal graph). Our study’s main goal is i) to highlight how our framework informs real-life settings and ii) validate and **complement** our theoretical insights. In that sense, we believe that our study serves its intended purpose. While this is a single dataset, to the best of our knowledge, it is the **first of its kind** in the strategic classification literature (which is mostly theoretical). A more detailed empirical investigation is an important future direction.
That said, we highlight that reviewer sN5p noted that *it is perfectly acceptable for a theory work to have a minimal empirical section*. We believe that **our theoretical contribution in itself, constitutes a significant standalone contribution, by bridging the gap between strategic classification theory and practice**.

---

### Decision · Program_Chairs · 2025-09-17

**Decision:**

Accept (poster)

**Comment:**

The paper studies strategic behavior by agents who can modify their features in settings where binary decisions are made.  Such agents may wish to invest effort modify their features to obtain a better outcome for themselves.  Such investment may also provide benefit to the principal. The resulting game has complex behavior. The paper analyzes this behavior, extracts insights, and presents a numerical example.

Strengths
- Interesting and novel setting
- Reasonable model
- Theory is strong

Weakness
- Reviewers were mixed on the relevance to practice. The paper does contain a numerical study of cardiovascular disease (e.g., Mw5J writes "the cardiovascular disease application is relevant and realistic.") but reviewers felt that a tighter connection to reality would help the paper

- Reviewers oibv and sN5 expressed concerns about real-world details not present in the model (e.g., the classifiers are linear). On the one hand, as I see it, the paper studies a parsimonious model in which (we hope) the core interesting dynamics of reality are also present. While some real-world complications are not modeled, this enables core insights to be extracted. This might not be possible in a model weighed down by more detail. On the other hand, if the model could handle more aspects of the real world, that would be better.

- Reviewers expressed concerns that the paper is dense.  I did not find this to be an extreme problem but I see where they are coming from.  The page limit at NeurIPS can present challenges to a paper like this.

- Some concerns about gaps in the literature review that can be addressed in a final version.

I am recommending acceptance because I feel the strengths outweight the weaknesses.